# Photocatalytic C-N coupling from stable and transient intermediates for gram-scale acetamide synthesis

Xin Li[1], Weiping Yang[1], Junping Yue[2], Jieyuan Li ®[1] ✉, Shujie Shen[1], Ruimin Chen[1], Jielin Wang[1], Huimin Dan[1], Dagang Yu ®[2] & Fan Dong[1] ✉

Electro/photocatalytic C-N coupling acts as a key build-block to the next generation of chemicals like amides for wide applications in energy, pharmaceuticals and chemical industries. However, the uncontrolled intermediates coupling challenges the efficient amide production regarding yield or selectivity. Here we propose a photocatalytic radical addition route, where the fundamental active species, including oxygen and photogenerated electron-hole pairs, are regulated for selective intermediates generation and efficient acetamide synthesis from mild co-oxidation of $CH_3CH_2OH$ and $NH_3$. Sufficient $CH_3CH_2OH$ is provided to accumulate the stable intermediate ($CH_3CHO$). Meanwhile, the limited $NH_3$ concentration ensures the controllable generation and fast addition of the transient radical ($^\bullet NH_2$) on $CH_3CHO$. Through the directed coupling of stable-transient intermediates, the acetamide synthesis rate is pushed forward to a hundred-mmol level ($105.61 \pm 4.86$ mmol·$g_{cat}^{-1}$·$h^{-1}$) with a selectivity of $99.17\% \pm 0.39\%$, reaching a gram-scale yield (1.82 g) of acetamide. These results illuminate valuable opportunities for the photocatalysis-driven synthetic industry.

Amides and their derivatives are some of the most important essential chemicals in the food, pharmacy, and chemical industries[1-3]. The efficient and economical synthesis of amides contributes greatly to the sustainable development of human society. However, industrial amide synthesis still relies on coupling the C- and N-containing small molecules under harsh conditions. Taking acetamide synthesis as a typical case, $CH_3COOH$ and $NH_3$ are fed for coupling with the assistance of dehydrators and catalysts at a high temperature of 150–180 °C, which consumes massive fossil fuels and leads to huge carbon emissions[4-6]. Hence, substantial research efforts have been devoted to developing alternative routes for amide synthesis under ambient conditions. Among them, the electrochemical method for co-reducing $CO_2/CO$ and $N_2/NO_x^-$, has recently attracted increased interest, in which obvious advances have been achieved for electrochemical amide synthesis from C-N coupling[7-11]. However, the corresponding synthesis

rate and selectivity (or Faraday efficiency, FE) are still restrained to low levels[12-14]. Specifically, under lower bias voltage, the activation of the reactant molecules is insufficient for their catalytic transformation, resulting in a limited synthesis rate[15-17]. On the contrary, the separate deep reduction of C- and N-substance is inevitable when the electric energy supply is excess, making it rather difficult to form the C-N bond selectively, thus failing to increase the selectivity of amides[18-20]. Hence, a mild redox ability, contributed by heterogeneous photocatalysis, may be an alternative approach for the efficient synthesis of amides from selective C-N coupling[21-24].

To this end, the precise regulation and selective generation of the key intermediates for C-N coupling, including carbonyl (-C=O) and amino (-NH₂) species, are imperative. However, these species are located in their intermediated valence states, respectively. Therefore, a precisely tailored mild redox reaction route should be constructed for

[1]Research Center for Carbon-Neutral Environmental & Energy Technology, Institute of Fundamental and Frontier Sciences, University of Electronic Science and Technology of China, Chengdu 611731, China. [2]Key Laboratory of Green Chemistry & Technology of Ministry of Education, College of Chemistry, Sichuan University, Chengdu 610064, China. ✉e-mail: jieyli@uestc.edu.cn; dongfan@uestc.edu.cn

their delicate preparation[25–27]. Otherwise, side reactions of the per-oxidation/per-reduction are unavoidable under a rough redox driving force, which results in decreased selectivity for amide synthesis. Moreover, it is well-acknowledged that a low reaction rate is expected by starting the coupling reactions with two stable intermediates due to their low chemical activity[28–30]. In contrast, the coupling possibility between two transient intermediates, such as reactive radicals, is severely limited since multiple disordered side reactions proceed in their short lifetimes[29,31,32]. That is, a complementary couple of stable-transient intermediates is required to endow the C-N coupling reaction with both optimum production rate and selectivity synergistically. By making use of the mild redox ability of photocatalysis and optimizing the coupled intermediates, it is anticipated that a meticulously designed photocatalytic reaction system can be developed for efficient synthesis of amides.

Herein, we propose a photocatalytic radical addition route for acetamide synthesis (Fig. 1), using $CH_3CH_2OH$ and $NH_3$ as the C- and N-source, respectively. Under the mild photocatalytic redox driving force, the fundamental active species, including $O_2$ and electron ($e^-$)-hole ($h^+$) pairs, are regulated for selective co-oxidation of $CH_3CH_2OH$ and $NH_3$. Specifically, sufficient $CH_3CH_2OH$ is provided to accumulate $CH_3CHO$ as the stable intermediate. Meanwhile, the limited provision of $NH_3$ ensures the controllable generation of $^\bullet NH_2$ as the transient intermediate, in which the fast addition of $^\bullet NH_2$ radical into $CH_3CHO$ is subsequently achieved. The oxidative ability of this reaction system is tailored by quantified $O_2$ activation, which guarantees considerable $CH_3CONH_2$ selectivity and impedes the peroxidation of $CH_3CHO$ and $^\bullet NH_2$ intermediates. The acetamide synthesis rate is pushed forward to a hundred-mmol level ($105.61 \pm 4.86$ mmol $g_{cat}^{-1}$ $h^{-1}$) with superior selectivity ($99.17\% \pm 0.39\%$), significantly exceeding most of the other catalytic routes. The photocatalysis system can be stably and continuously operated for 300 h, during which the acetamide yield reaches the gram scale (1.82 g). A comprehensive mechanism investigation is conducted to reveal the actual coupling coordinates and key intermediates, in which direct evidence has been provided based on the iso-type (D and $^{15}N$) labeled in situ EPR, HR-MS, NMR, and in situ ATR-FTIR. It is clarified that the efficient coupling of the stable and transient intermediates is the decisive factor for achieving optimum catalytic activity. We anticipate the current demonstration to be a starting point for realizing the efficient production of amide compounds with a simple but practical heterogeneous photocatalysis scheme, thus increasing the possibility of the science and industry revolution driven by solar light.

## Results and discussion

### Reaction system design and efficiency evaluation

The commercial $TiO_2$ (P25) is applied as a representative photocatalyst to develop the photoredox route for C-N coupling, which only absorbs the photon and forms the active $e^-$ and $h^+$ in this system, excluding the specificity of other modified photocatalysts and establishes a general pattern for the superiority of the photocatalytic amides synthesis scheme. Other common photocatalysts are also investigated in the C-N coupling reaction, which processes similar properties, showing the universality of photocatalysts in C-N coupling (Supplementary Fig. 4 and Supplementary Note 3). At first, various photocatalytic routes, including co-oxidation, co-reduction, and combined redox reactions, are screened to examine the practicability of amide synthesis (Table 1). It is identified that the amide bonds can be constructed through the coupling of various $C_1/C_2$ and $N_1$ substances, in which significant promotion of the synthesis efficiency is observed in the reaction route of $CH_3CH_2OH$ and $NH_3$ co-oxidation for $CH_3CONH_2$ production (109.24 μmol $h^{-1}$), remarkably exceeding the other photocatalysis routes. Therefore, $CH_3CH_2OH$ and $NH_3$ reactants are assigned as the respective C- and N-source for the comprehensive efficiency evaluation and mechanism investigations.

After the optimization of the reaction parameters, including $NH_3$ concentration, $CH_3CH_2OH$ proportion, $O_2$ proportion (in Ar), and catalyst dosage (Supplementary Figs. 5–9 and Supplementary Note 16), a high $CH_3CONH_2$ production rate is achieved at $105.61 \pm 4.86$ mmol $g_{cat}^{-1}$ $h^{-1}$ (Fig. 2a) from the co-oxidation of $CH_3CH_2OH$ and $NH_3$, which is a significant progression as it reaches the hundred-mmol level for photocatalytic C-N coupling under ambient conditions. Then, as the complete redox reaction proceeds in a heterogeneous photocatalysis process, it is deduced that the co-oxidation reaction is contributed by the light-generated $h^+$ and $O_2$-enabled active species, in which the detailed mechanism requires further investigation. Meanwhile, the hydrogen gas ($H_2$) is yielded by the cooperative $e^-$-driven reduction reaction with the dehydrogenated hydrogen from $CH_3CH_2OH$ and $NH_3$ (Supplementary Note 13 and Supplementary Figs. 10, 11), in which a minor amount of $H_2$ is detected due to competing $O_2$ activation and reduction for $e^-$ consumption.

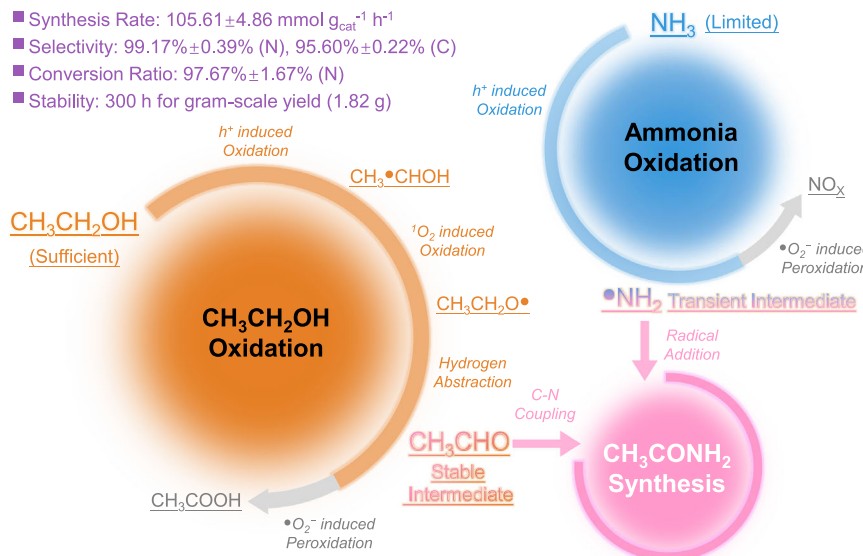

**Fig. 1 | Schematic diagram.** Illustration for the coupling of stable and transient intermediates for acetamide photosynthesis.

Subsequently, the $CH_3CONH_2$ selectivity is evaluated, corresponding to $NH_3$ (N) and $CH_3CH_2OH$ (C), respectively (Supplementary Note 15). As depicted in Fig. 2b and Supplementary Figs. 12–20, the concentration of $CH_3CONH_2$ gradually increases along with the consumption of $NH_3$. It is noted that trace $N_2$ byproduct is generated at the first 30 min, which originates from the $NH_3$ peroxidation. However, as the reaction proceeds, the accumulation of $N_2$ is impeded due to the decrease of the $NH_3$ concentration, which illustrates that the controllable provision of the initial $NH_3$ concentration is vital to determine

the oxidative pathways of the system. The concentrations of the other byproducts are exceedingly low, resulting in data points in Fig. 2b that are so closely aligned they appear to overlap, making individual distinctions hard to discern. Under the limited provision of $NH_3$ (100.00 mg $L^{-1}$), a near-complete conversion of $NH_3$ (97.67% ± 1.67%) for $CH_3CONH_2$ production is accomplished under the 60 mins' light irradiation, presenting a superior $CH_3CONH_2$ selectivity (N) of 99.17 ± 0.39%. In addition, it is observed that the synthesis rate decreases after 45 min, which is reasonable due to the remaining $NH_3$ concentration being low after rapid consumption. Based on these results, the recovery of the reaction rate and maintaining of selectivity are expected by conducting cycled experiments through periodical input of limited $NH_3$.

Then, the $CH_3CONH_2$ selectivity regarding $CH_3CH_2OH$ is also evaluated by comprehensively detecting the oxidative products in both the liquid and gaseous phases (Fig. 2c and Supplementary Figs. 21–25). The corresponding selectivity is ~100% at the first 30 min. Along with the consumption of $NH_3$, the generation of the trace byproduct ($CH_3COOH$) is observed due to the peroxidation of $CH_3CH_2OH$ under low $NH_3$ concentration, which again verifies the importance of limited $NH_3$ provision to guarantee the superior selectivity. After the photocatalysis reaction for 1 h, the $CH_3CONH_2$ selectivity (C) is maintained at as high as 95.60 ± 0.22%. The reaction parameters of $NH_3$ concentration for the $CH_3CONH_2$ selectivity test regarding $NH_3$ and $CH_3CH_2OH$ are set differently. Specifically, a lower concentration of $NH_3$ (100.00 mg $L^{-1}$) is provided for its efficient conversion to evaluate the selectivity (N). While more $NH_3$ (800.00 mg $L^{-1}$) is included to impede the peroxidation of $CH_3CH_2OH$ after the rapid $NH_3$ consumption, where objective evaluations can be established for the selective oxidation of $NH_3$ and $CH_3CH_2OH$ for $CH_3CONH_2$ synthesis, respectively (Supplementary Note 17). These selectivity results imply that the key C- and N-intermediate must be

**Table. 1 | List of screening experiments of different photocatalytic redox routes for amide synthesis**

| C-source | N-source | Product | Synthesis rate (μmol h$^{-1}$)$^a$ |
|---|---|---|---|
| $CH_3OH$ | $NH_3$ | $HCONH_2$ | 51.99 |
| $CH_3OH$ | $NO_2^-$ | $HCONH_2$ | 40.44 |
| $CH_3OH$ | $NO_3^-$ | $HCONH_2$ | 41.21 |
| $HCOOH$ | $NH_3$ | $HCONH_2$ | 0.32 |
| $HCOOH$ | $NO_2^-$ | $HCONH_2$ | trace |
| $HCOOH$ | $NO_3^-$ | $HCONH_2$ | 0.57 |
| $CH_3CH_2OH$ | $NH_3$ | $CH_3CONH_2$ | 109.24 |
| $CH_3CH_2OH$ | $NO_2^-$ | $CH_3CONH_2$ | 23.41 |
| $CH_3CH_2OH$ | $NO_3^-$ | $CH_3CONH_2$ | 16.01 |
| $CH_3COOH$ | $NH_3$ | $CH_3CONH_2$ | trace |
| $CH_3COONa$ | $NH_3$ | $CH_3CONH_2$ | trace |
| $CH_3COOH$ | $NO_2^-$ | $CH_3CONH_2$ | 6.55 |
| $CH_3COOH$ | $NO_3^-$ | $CH_3CONH_2$ | trace |

$^a$The respective standard curves of $HCONH_2$, $CH_3CONH_2$, and $CO(NH_2)_2$ (Supplementary Figs. 1–3) are provided in the Supplementary Information.

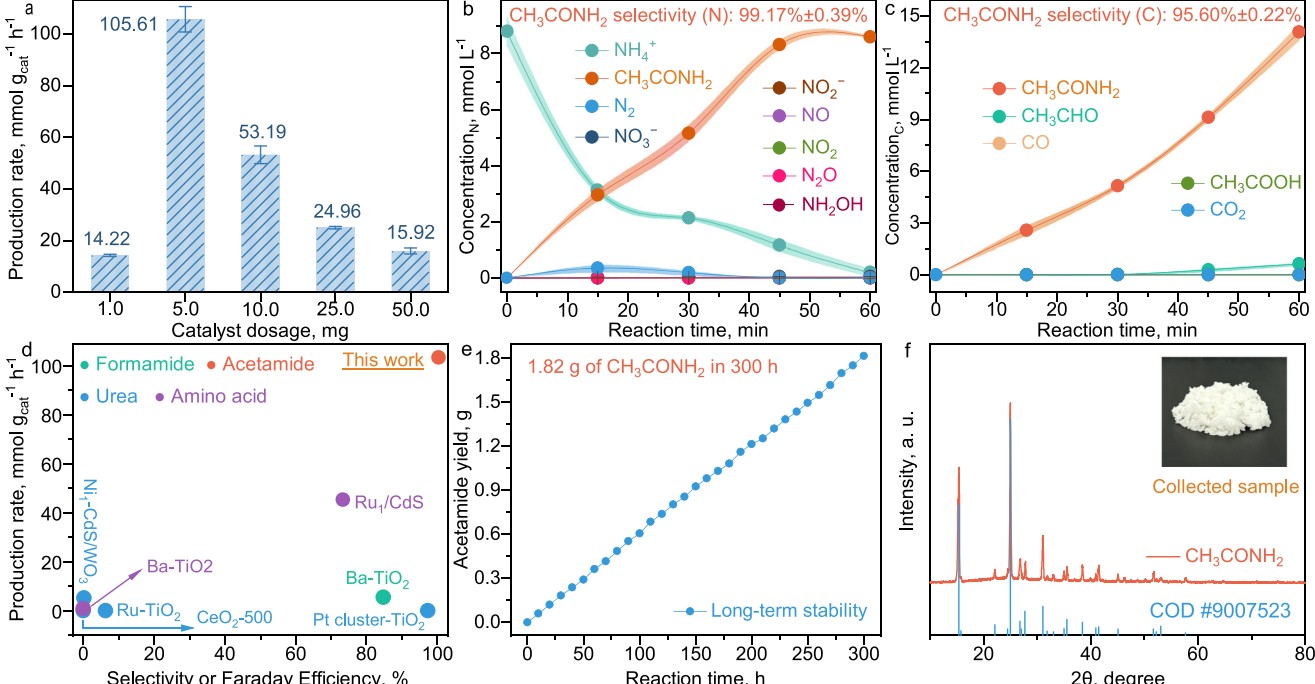

**Fig. 2 | Efficiency evaluation for acetamide photosynthesis. a** catalyst dosage-dependent unit production rate; $CH_3CONH_2$ selectivity evaluation regarding the N- (**b**) and C-sources (**c**), respectively; **d** Efficiency comparison between different catalytic routes for photocatalytic C-N coupling, including the targets of production rate and selectivity, the corresponding research works are listed and cited in Supplementary Information (Supplementary Table 3)[54–60]; **e** Long-term stability test. **f** XRD pattern and the image (inset) of the collected $CH_3CONH_2$ sample with rotary evaporation after the long-term stability test. The respective standard curves for detecting the reaction species using ion chromatography (IC, $NH_4^+$, $NO_2^-$, $NO_3^-$, $CH_3COOH$, and $NH_2OH$, Supplementary Figs. 31–35) are provided in Supplementary Information. The error bars in (**a–c**) were drawn based on the calculated standard error of two parallel tests.

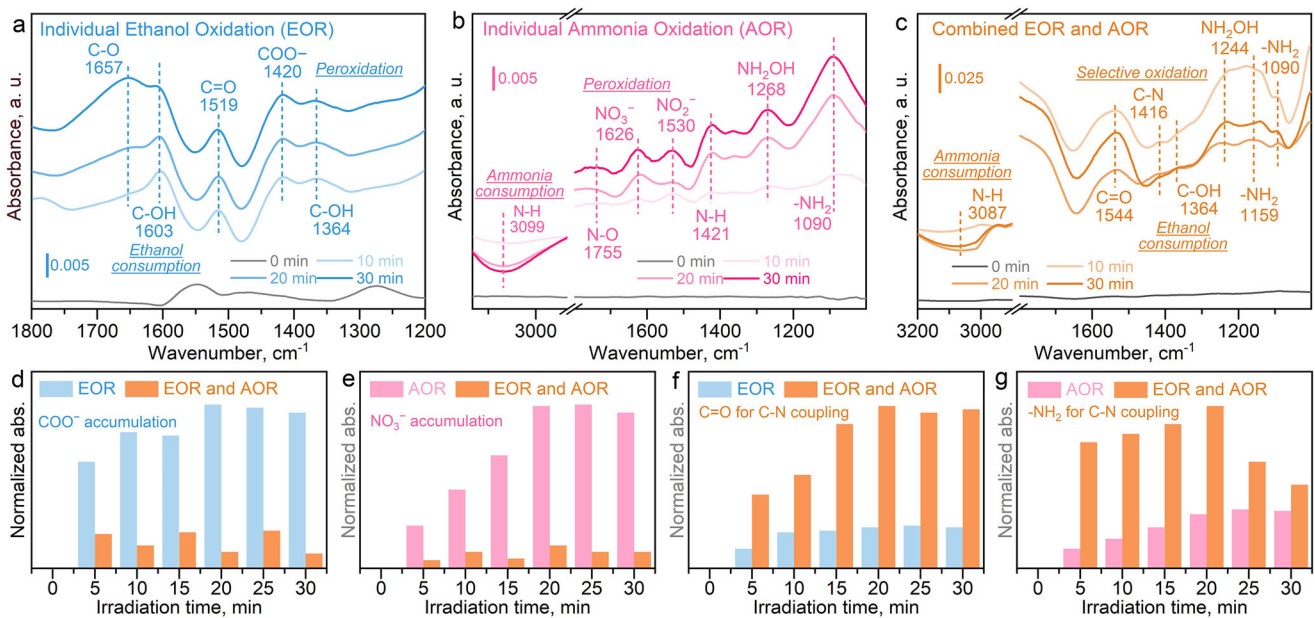

**Fig. 3 | In situ ATR-FTIR investigation for revealing the reaction coordinates.** Time-dependent IR signals for the individual ethanol oxidation reaction (EOR, **a**), ammonia oxidation reaction (AOR, **b**), and combined EOR and AOR (**c**), respectively; Normalized results of the IR signals of COO⁻ (**d**), NO₃⁻ (**e**), C=O (**f**), and -NH₂ (**g**), respectively. The full spectra of the adsorption and photocatalysis process are provided in Supplementary Information for individual EOR (Supplementary Fig. 37), individual AOR (Supplementary Fig. 38), and combined EOR and AOR (Supplementary Fig. 39), respectively.

clarified and optimized for the selective C-N coupling, which will, in turn, impede the individual peroxidation of $CH_3CH_2OH$ and $NH_3$. Moreover, when compared with the conventional industrial synthesis route (Supplementary Table 2), the photocatalytic co-oxidation route for $CH_3CONH_2$ synthesis demonstrates significant progress in terms of synthesis conditions, raw material price, and synthesis efficiency.

After systematically optimizing the reactants and reaction parameters, the establishment of the highly effective and selective co-oxidation route of $CH_3CH_2OH$ and $NH_3$ lays the theoretical foundation for developing the photocatalytic $CH_3CONH_2$ synthesis route. In comparison with the other reported routes for photocatalytic C-N coupling (Fig. 2d and Supplementary Table 3), it is significant that a high efficiency is accomplished by applying the photocatalytic co-oxidation routes, including the important indexes of synthesis rate of the targeted coupling products and the corresponding selectivity. Moreover, the as-constructed reaction system has been continuously operated with sufficient $CH_3CH_2OH$ provision for 300 h (Supplementary Table 1), in which periodic $NH_3$ conversion for stable $CH_3CONH_2$ production is maintained, delivering a gram scale yield of $CH_3CONH_2$ (1.82 g, Fig. 2e). The collected solid sample, with rotary evaporation for its recovery, is verified as $CH_3CONH_2$ by the XRD and nuclear magnetic resonance (¹H NMR) technologies, consistent with the crystallography open database (COD #9007523, Fig. 2f) and chemical shift of the hydrogen atom (Supplementary Fig. 26) respectively. No deactivation of the P25 is observed based on the characterization results of the sample before and after the long-term stability test (Supplementary Figs. 27–30 and Supplementary Note 18).

**Reaction coordinates for the selective oxidation of $CH_3CH_2OH$ and $NH_3$**
To identify the critical step of the co-oxidation of $CH_3CH_2OH$ and $NH_3$. The variate-controlled in situ attenuated total reflection Fourier transform infrared spectroscopy (ATR-FTIR, Supplementary Fig. 36) investigation is conducted by comparing the IR signals for individual ethanol oxidation (EOR), ammonia oxidation (AOR) and their co-oxidation reactions. As the individual EOR proceeds (Fig. 3a, the spectra for the adsorption equilibrium process in the dark are

displayed in Supplementary Fig. 37 and Supplementary Table 4), the gradual consumption of $CH_3CH_2OH$ (C-OH at 1603 and 1364 cm⁻¹) is observed[33], which generates the corresponding oxidative intermediates (C-O at 1657 and C = O at 1519 cm⁻¹)[34,35]. It is found that the peroxidative product (COO⁻ at 1420 cm⁻¹)[36] is produced by the excess oxidative ability in the individual EOR. Similarly, the consumption of $NH_3$ (N-H at 3099 cm⁻¹)[37] yields the intermediated products of -NH₂ (1090 cm⁻¹)[38] and $NH_2OH$ (1268 cm⁻¹)[8] in the individual AOR (Fig. 3b, the spectra for the adsorption equilibrium process in the dark are displayed in Supplementary Fig. 38 and Supplementary Table 5), in which the peroxidative products are also observed (N-O at 1755 cm⁻¹, $NO_2^-$ at 1530 cm⁻¹ and $NO_3^-$ at 1626 cm⁻¹)[23,39]. Since $NH_2OH$ species are not detected during the reaction process (Fig. 2b, Supplementary Note 14, and Supplementary Fig. 20), it is clarified that $NH_2OH$ is not desorbed from the catalyst surface but is instead converted to other intermediates and products.

Most importantly, as illustrated in the combined co-oxidation reactions of EOR and AOR (Fig. 3c, the spectra for the adsorption equilibrium process in the dark are displayed in Supplementary Fig. 39 and Supplementary Table 6), as the co-oxidation of $CH_3CH_2OH$ (C-OH at 1364 cm⁻¹) and $NH_3$ (N-H at 3087 cm⁻¹) proceeds, the interaction of the C- (C=O at 1544 cm⁻¹) and N-intermediates (-NH₂ at 1159 and 1090 cm⁻¹) leads to the generation of the coupling products as indicated by the C-N bond formation at 1416 cm⁻¹ [40]. Therefore, it is identified that the amide bond (-CONH₂) can be effectively generated through the coupling of C=O and -NH₂-based intermediates. However, their exact structures require further investigation.

Then, the signals for the peroxidative byproducts (Fig. 3d, e) and critical coupling intermediates (Fig. 3f, g) are normalized for comparison between the individual and combined oxidation reactions. The detailed methods for normalization of the FTIR signals are presented in Supplementary Note 8. It is observed that more peroxidative byproducts, including COO⁻ from EOR (Fig. 3d) and $NO_3^-$ from AOR (Fig. 3e), are accumulated in the respective individual oxidation reactions, which hinders the selective generation of key intermediates for C-N coupling. By the combination of EOR and AOR, the intensified signals for the intermediates of C=O (Fig. 3f) and -NH₂ (Fig. 3g) are

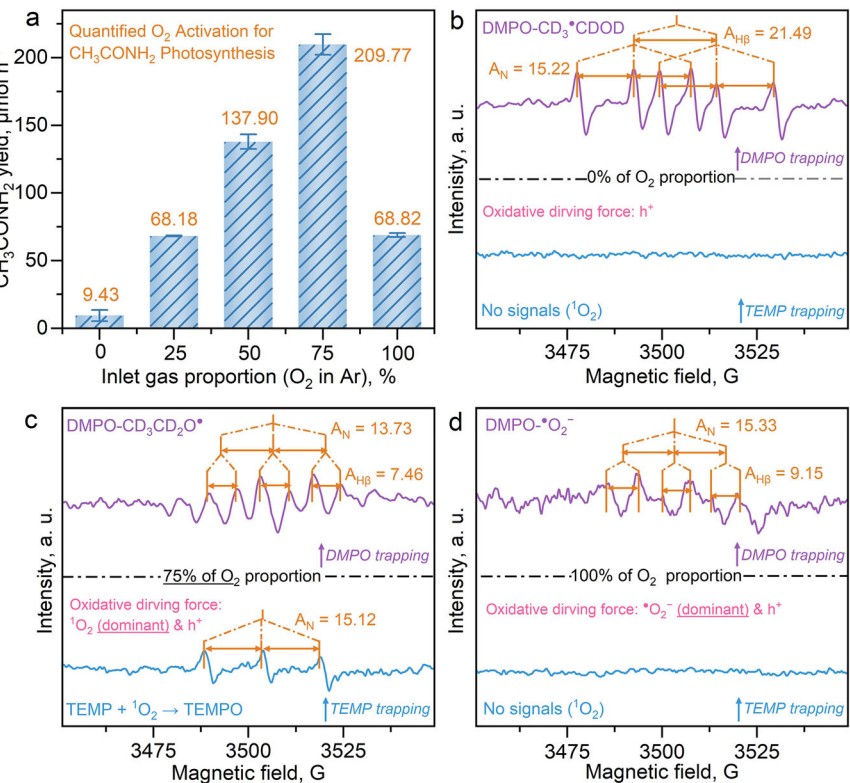

**Fig. 4 | The mechanisms of O₂ regulation for generating the coupling intermediates. a** O₂ proportion-dependent CH₃CONH₂ yield; DMPO-trapping (with D-labeling, images above) and TEMP-trapping (images below) in situ EPR experiments under the O₂ proportion of 0% (**b**), 75% (**c**), and 100% (**d**) respectively. The error bars in (**a**) were drawn based on the calculated standard error of two parallel tests.

noted, leading to the directed coupling rather than the individual peroxidation. Furthermore, since the $CH_3CH_2OH$ is sufficiently provided, the continuous accumulation of C=O species is reasonable, which provides sufficient C-intermediate for coupling. On the contrary, the limited provision of $NH_3$ yields -$NH_2$ at the first 20 min. Then -$NH_2$ is consumed along with the decrease of $NH_3$ concentration, which follows the efficiency test result (Fig. 2b). Hence, it is deduced that the -$NH_2$-based species can be identified as the key N-intermediate which possesses higher reactive activity than that of the C=O species, due to its gradual accumulation and rapid consumption. It is again verified that continuous and effective C-N coupling can be accomplished by providing sufficient $CH_3CH_2OH$ and periodically supplying limited $NH_3$. This approach not only generates C- and N-intermediates for coupling but also avoids their peroxidation.

**Generation and coupling mechanism of the key intermediates**
Based on the in situ ATR-FTIR investigation, it is acknowledged that the directed coupling of the C- and N-intermediates is vital to selectively producing $CH_3CONH_2$. Then, the generation pathways and coupling mechanism of these key intermediates should be revealed. The $O_2$-proportion-dependent $CH_3CONH_2$ synthesis efficiency is first evaluated to identify the oxidative driving force for the generation of these intermediates (Fig. 4a and Supplementary Figs. 40, 41). It is observed that the $CH_3CONH_2$ yield increases along with the elevation of $O_2$ proportion (in Ar) from 0% to 75%, reaching a yield of 209.77 ± 7.68 μmol h⁻¹. However, a rapid decrease of $CH_3CONH_2$ yield (68.82 ± 1.56 μmol h⁻¹) is noted under excess $O_2$ provision (100%), which directly confirms that the $O_2$ molecules, as fundamental active species, require to be quantificationally provided for mild $O_2$ activation, which enables the precise generation of the coupling intermediates. In addition, the production of e⁻-driven $H_2$ is elevated at the $O_2$ proportion of 0% than that of 75%, which again confirms that the $O_2$

activation and reduction work as competing reactions with $H_2$ evolution (Supplementary Fig. 42).

Then, a comprehensive iso-type (deuterium, D) labeled in situ electron paramagnetic resonance (EPR) experiment is conducted to dynamically track the selective EOR pathways and corresponding intermediates, by applying the 5,5-dimethyl-1-pyrroline *N*-oxide (DMPO) and 2,2,6,6-Tetramethylpiperidinooxy (TEMPO) as the respective trapping agents under different $O_2$ proportion (Fig. 4b–d)[41–43]. The $CD_3CD_2OD$, rather than the other common reagents such as $CH_3OH$ or $H_2O$, is applied as the only solvent for the D-labeled in situ EPR tests to precisely reproduce the selective EOR for acetamide synthesis. Under the 0% of $O_2$ proportion, it is found that the EOR proceeds under the oxidative driving force of h⁺, as evidenced by the successful detection of the oxidative intermediate of the DMPO trapped alkyl radical (DMPO-$CD_3$•CDOD) and the absence of the $O_2$-enabled active species (Fig. 4b and Supplementary Figs. 43–45). Interestingly, DMPO trapped alkoxy radical (DMPO-$CD_3CD_2O$•) and trace DMPO-$CD_3$•CDOD are both detected under the condition of an $O_2$ proportion of 75%, which follows the optimum $O_2$ proportion parameter for acetamide synthesis (Fig. 4c, Supplementary Fig. 46, 47, and Supplementary Note 19), illustrating the direct relationship between $CD_3CD_2O$• generation and acetamide yield promotion. There is a tautomerization transformation from $CD_3$•CDOD to $CD_3CD_2O$•, which is driven by the mild oxidative ability of singlet oxygen (¹O₂, Fig. 4c and Supplementary Fig. 48). The ¹O₂ is generated from either the activation of the triplet oxygen (³O₂) or the h⁺-triggered rapid oxidation of superoxide radical (•O₂⁻)[44,45]. Since ³O₂ and •O₂⁻ are difficult to distinguish in the current in situ EPR experiments, it is reasonable to infer that both of these two reaction pathways are involved for the generation of ¹O₂. Moreover, the generation of $CD_3CD_2O$•, which is an O-centered active radical, implies that the C-OD bond in the $CD_3CD_2OD$ molecule can be effectively activated by dehydrogenation,

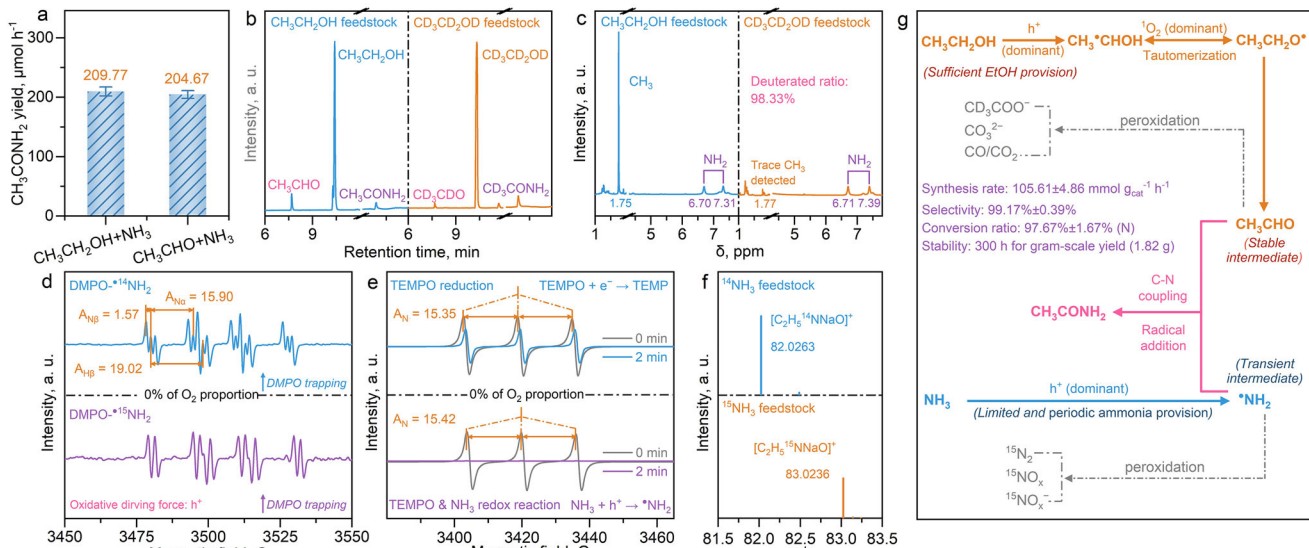

**Fig. 5 | Transformation and coupling mechanism for the C- and N-intermediates. a** Control experiments for CH$_3$CONH$_2$ synthesis by replacing CH$_3$CH$_2$OH with CH$_3$CHO as the C-source; GC (**b**) and $^1$H NMR (**c**) results with D- (images right) and H-labeling (images left) respectively; **d** DMPO-trapping in situ EPR experiments with $^{14}$N- (image above) and $^{15}$N-labeling (image below) respectively; **e** in situ EPR spectra for e$^-$-induced TEMPO consumption with (image below)

and without (image above) NH$_3$ provision respectively; **f** HR-MS results for CH$_3$CONH$_2$ detection with $^{14}$N- (image above) and $^{15}$N-labeling (image below) respectively; **g** Proposed reaction pathways of CH$_3$CH$_2$OH and NH$_3$ co-oxidation for intermediates coupling and CH$_3$CONH$_2$ synthesis. The error bars in (**e**) were drawn based on the calculated standard error of two parallel tests.

which yields the precursor of -C=O species for C-N coupling. As the concentration of O$_2$ increases, the signal intensity of $^\bullet$O$_2^-$ radical gradually increases while the signal intensity of $^1$O$_2$ gradually decreases. Finally, the excess provision of O$_2$ (100%) leads to the accumulation of the strong oxidative $^\bullet$O$_2^-$ radicals (Fig. 4d, Supplementary Fig. 49, and Supplementary Note 20), in which no signals of the $^1$O$_2$ are detected (Fig. 4d and Supplementary Figs. 50, 51). As the $^\bullet$O$_2^-$ radicals bind to the reactants to generate peroxidative products, $^1$O$_2$ species are no longer generated. Hence, the $^\bullet$O$_2^-$-induced peroxidation is inevitable under such a high O$_2$ concentration, resulting in the decreased coupling efficiency.

By the quantified O$_2$ activation at the proportion of 75%, it is confirmed that the peroxidation of CD$_3$CD$_2$O$^\bullet$ is avoided. Therefore, it is reasonably deduced that stable C-intermediate of acetaldehyde (CH$_3$CHO), which possesses the critical -C=O group, can be generated through the hydrogen abstraction from CD$_3$CD$_2$O$^\bullet$. The corresponding control experiment is thereby conducted by replacing CH$_3$CH$_2$OH with CH$_3$CHO as the initial reactant for C-N coupling (Fig. 5a, Supplementary Note 10, Supplementary Fig. 52, and Supplementary Table 7). It is noted that almost complete recovery of the CH$_3$CONH$_2$ yield is accomplished via the NH$_3$-CH$_3$CHO condensation (204.67 ± 6.58 μmol h$^{-1}$), consistent with that of the co-oxidation of NH$_3$ and CH$_3$CH$_2$OH (209.77 ± 7.68 μmol h$^{-1}$), which identifies the CH$_3$CHO as the stable C-intermediate, where the generation of CH$_3$CHO requires precise regulation to avoid the decrease of the acetamide. Furthermore, the key role of CH$_3$CHO is illustrated by the gas chromatography (GC, Fig. 5b) and corresponding mass spectra (MS, Supplementary Figs. 53, 54 and Supplementary Note 21) detection results. The provision of CH$_3$CH$_2$OH/CD$_3$CD$_2$OD feedstock results in the generation of H/D labeled acetaldehyde and acetamide, respectively, presenting a selective EOR along the route of CH$_3$CH$_2$OH → CH$_3$$^\bullet$CHOH ↔ CH$_3$CH$_2$O$^\bullet$ → CH$_3$CHO → CH$_3$CONH$_2$.

Also, it is clarified by the $^1$H NMR (Fig. 5c, Supplementary Figs. 55 and 56 and Supplementary Note 22) results that the CD$_3$CONH$_2$ is detected with a deuterated ratio of 98.33%, which verifies the reliability and practicability of these designed D-labeled in situ experiments.

After successfully identifying the EOR for accumulating the stable C-intermediate of CH$_3$CHO, the cooperative AOR mechanism is revealed by a series of $^{15}$N-/$^{14}$N-labeled tracking experiments. The DMPO-$^{\bullet 14}$NH$_2$ and DMPO-$^{\bullet 15}$NH$_2$ are detected without O$_2$ provision, respectively (Fig. 5d, Supplementary Figs. 57, 58, and Supplementary Note 23)[46,47], manifesting that h$^+$ is the dominant driving force for the mild oxidation of NH$_3$ for $^\bullet$NH$_2$ generation. By combining the in situ ATR FTIR and in situ EPR spectra (Fig. 3c, g, 4h), it is indicated that the absorbed -NH$_2$ is oxidized by h$^+$ to produce $^\bullet$NH$_2$, which is involved in subsequent C-N coupling processes. Besides, further in situ EPR experiments are applied by using TEMPO as the indicator. Under the condition of individual TEMPO provision (Fig. 5e and Supplementary Fig. 59), a slight decrease of TEMPO signals is recorded due to the TEMPO reduction driven by e$^-$ under light irradiation for 2 min On the contrary, near-complete TEMPO consumption is observed when adding NH$_3$ into the test solution (Fig. 5e and Supplementary Fig. 60). Since no other oxidative driving force is included due to the absence of O$_2$ (proportion at 0%), it is again confirmed that the selective AOR is dominantly driven by h$^+$ for $^\bullet$NH$_2$ generation, which consumes more h$^+$ and in turn provides extra e$^-$ for accelerating TEMPO reduction.

By the combination of the efficiency test (Figs. 2b, 4a), the in situ ATR-FTIR results (Fig. 3c), as well as the $^{15}$N-labeled in situ EPR investigations for $^\bullet$NH$_2$ detection (Fig. 4h, i), it is acquired that no peroxidative N-containing intermediates or products are generated in the combined EOR and AOR under quantified O$_2$ activation, which leads to the ~100% selectivity (N) for CH$_3$CONH$_2$ synthesis. Hence, it is concluded that the $^\bullet$NH$_2$ is identified as the transient N-intermediate for C-N coupling. Since the reaction rate coefficient of NH$_3$-to-NH$_2$ oxidation is an order of magnitude faster than that of the NH$_2$-to-NH oxidation, the selective oxidation of NH$_3$ primarily yields NH$_2$-based intermediates or radicals than that of the NH[48,49]. Moreover, the fast formation (~1 ps) and relatively long lifetime (>1 ns) endow the $^\bullet$NH$_2$ radicals with a high possibility to participate in the coupling reactions, in comparison with the other N-centered radicals possessing short lifetimes, such as $^\bullet$NH$_4$ (13 ps)[50,51], which again verifies that $^\bullet$NH$_2$ can act as the transient N-intermediate for C-N coupling. The illustration for $^\bullet$NH$_2$ detection parameters, especially the O$_2$ proportion, is listed

in Supplementary Note 24, which explains that the absence of the $O_2$ provision is indispensable for the detection of $\bullet NH_2$. Based on these results, it is clarified that the controllable generation and fast addition of transient $\bullet NH_2$ radicals on stable $CH_3CHO$ intermediate should contribute directly to the efficient and selective C-N coupling for $CH_3CONH_2$ synthesis. The AOR pathways, along with the route of $NH_3 \rightarrow \bullet NH_2 \rightarrow CH_3CONH_2$, are then verified by the $^{14}N/^{15}N$ labeled high-resolution MS (HR-MS, Fig. 5f, Supplementary Figs. 61, 62, and Supplementary Note 25) results, in which both $CH_3CO^{14}NH_2$ and $CH_3CO^{15}NH_2$ are detected, by introducing $^{14}NH_3$ and $^{15}NH_3$ (in $CH_3CH_2OH$) as the feedstocks respectively.

After these comprehensive mechanism investigations of the reaction pathways and coupling intermediates, the heterogeneous photocatalytic scheme for $CH_3CONH_2$ synthesis is proposed (Fig. 5g). Under the sufficient provision of $CH_3CH_2OH$ and quantified $O_2$ activation (75% in Ar), the $CH_3CHO$, from selective oxidation of $CH_3CH_2OH$, is generated and accumulated as the stable C-intermediate for coupling. Meanwhile, the limited and periodic provision of $NH_3$ leads to the controllable generation and fast addition of transient N-intermediate ($\bullet NH_2$) on stable $CH_3CHO$ intermediate, in which the rapid consumption of $\bullet NH_2$ for C-N coupling avoids its peroxidation for side product generation. The coupling of long-lived C-intermediate and short-lived N-intermediate ensures a high selectivity and yield rate of $CH_3CONH_2$. The C-N coupling selectivity can be effectively maintained by the tailored oxidative ability and oxidative species, including $h^+$ and $e^-$-driven $O_2$-enabled active species ($^1O_2$ and $\bullet O_2^-$), in which the excess $e^-$ contributes to the $H_2$ evolution to construct a complete photocatalytic redox reaction. For the overall photocatalytic synthesis pathway of $CH_3CONH_2$, optimizing the interaction between the initial reactive species and the photogenerated charge carriers is of crucial importance to achieve carrier-driven radical generation. By fully leveraging the radical characteristics of lifetime and migration distance, radical-mediated photocatalytic reactions can proceed effectively, which ensures the high efficiency of mass transfer and the sufficient utilization of light energy, thereby achieving a significant enhancement of the redox efficiency.

Here, we have demonstrated a compelling solar-driven photocatalytic synthesis system. By combining mild co-oxidation of $CH_3CH_2OH$ and $NH_3$, efficient and selective access to $CH_3CONH_2$ synthesis is achieved with heterogeneous photocatalysis technology. Through the precise regulation of the $O_2$ activation and $e^-$-$h^+$ carriers, the $CH_3CONH_2$ yield reaches the gram scale with satisfying production rate, selectivity, and stability. Comprehensive mechanism investigation results indicate that the controllable generation and fast addition of the transient $\bullet NH_2$ radical on stable $CH_3CHO$ intermediate contributes essentially to this catalytic performance in acetamide synthesis. The C-N coupling from stable and transient intermediates could provide scientific and technical feasibility for the solar light-driven synthetic fields, which, with further promotion, might be industrially and economically practical, thereby illuminating valuable opportunities for the mild-photocatalysis-driven synthetic industry.

## Methods

### Photocatalytic synthesis route and efficiency evaluation
Different N-source ($NH_3$, $KNO_2$, or $KNO_3$) was added into 50.00 mL of absolute methanol ($CH_3OH$), ethanol ($CH_3CH_2OH$), formic acid (HCOOH), or acetic acid ($CH_3OOH$) to screen the C-N coupling routes (Supplementary Note 1). After the construction of the route of $CH_3CH_2OH$ and $NH_3$ co-oxidation for $CH_3CONH_2$ synthesis, the parameters of $NH_3$ concentration, $CH_3CH_2OH$ volume, $O_2$ proportion, and catalyst dosage were optimized for elevating the synthesis efficiency (Supplementary Note 2). The corresponding parameters of the selectivity test regarding N and C were also optimized (Supplementary Notes 4, 5). The reactants, intermediates, and products, in liquid and gaseous phases, were detected and quantified by high-performance liquid chromatography (HPLC, Shimadzu Essentia LC-16i/MSD), ion chromatography (IC, Shimadzu IC-16) and infrared flue gas analyzer (Bruker MATRIX-MG5), respectively (Supplementary Note 26).

After the long-term stability test, the collected catalyst sample was characterized by the X-ray diffraction technology (XRD, Shimadzu XRD-6100) for its crystal stability, scanning electron microscopy (SEM, FEG ESEM XL30) and transmission electron microscopy (TEM, FEI Talos F200S) for its geometric stability respectively. The obtained reaction mixture was centrifuged for the separation of the solution and catalyst. Then, a rotary evaporator was applied to remove the solvent and potential side products from the solution. The collected product sample was characterized by the XRD (Supplementary Notes 6, 7).

### In situ ATR-FTIR investigation
An INVENIO R FTIR (Bruker) spectrometer equipped with a mercury cadmium telluride (MCT) detector was utilized for the measurements. Before the test, 100.0 uL of catalyst ink (a mixture of 10.00 mg of P25, 25.0 uL of Nafion, and 1.00 mL of $CH_3CH_2OH$) was deposited onto the Si crystal and then dried in air. The reaction chamber was filled with a total of 10.00 mL of the reaction solution, and 75% of $O_2$ (in Ar) was continuously injected into the system. An Xe lamp (Bobei BBZM-1) was applied as the light source, and detection was performed during light irradiation[52].

The pristine IR signals were normalized to evaluate the species' evolution directly. Between the columns of data, the highest value was set to be 1, and the lowest value was set to be 0. The rest were correspondingly normalized from 0 to 1. The resulting normalized data were thus described as a function of the IR scanning time[53].

### In situ EPR investigation by H/D labeling
All the in situ EPR measurements were conducted on the equipment of Bruker EMX Nano. 1000.0 µL of $CH_3CH_2OH$ (or $CD_3CD_2OD$), 40.0 µL of $NH_3 \cdot H_2O$, 50.0 µL of the well-mixed P25 suspension (1000.00 mg $L^{-1}$ in $CH_3CH_2OH$ or $CD_3CD_2OD$), 10.0 µL of DMPO were added into the reactor under the $O_2$ proportion (in Ar) of 0, 75, and 100%, respectively. The EPR signals were recorded at 1, 3, 5, and 10 min under the dark or light irradiation conditions, respectively, collected by a capillary (Supplementary Notes 9).

### In situ EPR investigation by $^{14}N/^{15}N$ labeling
About 550.0 µL of $^{14}NH_3 \cdot H_2O$ (10.00 g $L^{-1}$) or $^{15}NH_4Cl$ (10.00 g $L^{-1}$ of $^{15}NH_4^+$), 390.0 µL of DI and 50.0 µL of the well-mixed P25 suspension (in $CH_3CH_2OH$, 1000.00 mg $L^{-1}$) were added into the reactor under the Ar (99.999%) gas injection. The EPR signals were recorded at 1, 3, 5, and 10 min under the dark or light irradiation condition respectively, collected by a capillary (Supplementary Notes 11, 12).

The other experimental details are provided in the Supplementary Information.

## Data availability
All source data generated in this study are provided in the Supplementary Information and Source Data files. Source data are provided with this paper.

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

## Acknowledgements

This work was supported by the National Natural Science Foundation of China (Grant Nos. 22225606, 22422607, and 22276029), the Young Elite Scientists Sponsorship Program by CAST (2023QNRC001) and the Natural Science Foundation of Sichuan Province (2025NSFTD0003).

## Author contributions

F.D. and J.L. conceived the idea and designed this study. X.L. conducted characterizations, photocatalytic tests, and mechanism investigations. W.Y. helped develop the photocatalytic C-N coupling reaction routes. J.Y. conducted the HR-MS and NMR detection experiments. S.S. helped screen the reaction routes from various C- and N-sources. R.C. and J.W. helped design the reactors and test the parameters for photocatalytic reactions and in situ characterizations. H.D. helped conduct the long-term stability test. D.Y. provided suggestions for conducting the mechanism investigation of intermediate coupling. X.L., J.L., and F.D. wrote the paper. All authors discussed the results and edited the paper.

## Competing interests

The authors declare no competing interests.
