## [Peer Review File · Nature Communications]

Photocatalytic C-N coupling from stable and transient intermediates for gram-scale acetamide synthesis

Corresponding Author: Professor Fan Dong

Version 0:

Reviewer comments:

Reviewer #1

(Remarks to the Author)

The manuscript authored by Li et al demonstrates that a couple of stable-transient intermediates generated during photocatalytic reactions can offer a high opportunity for C-N coupling. In their work, CH₃CH₂OH is used as the C-source, which undergoes oxidation into a stable intermediate, CH₃CHO; NH₃ is the N-source, the amount of which is limited so as to control the generation of a transient intermediate, ·NH₂. The fast coupling reaction between these two intermediates leads to a high CH₃CONH₂ synthesis rate and a promising reaction selectivity. This reaction route is innovative, and the in situ measurements have provided reliable evidence for both intermediates participating in the reaction pathways. The manuscript is publishable provided the following comments are addressed:

1. The discussion on the role of photocatalyst is missing in this manuscript. How does TiO₂ steer the oxidative reactions of both CH₃CH₂OH and NH₃? It would be best if the authors could provide some test for other photocatalysts and justify that the performances are similar. Otherwise, the catalyst characteristics (catalytic active sites) may also be a decisive factor.
2. Figs 3b and c indicate a markedly increased amount of NH₂OH species during the reactions, but it has not been displayed in Figs 2 and 4. A detailed evaluation of the role of NH₂OH should be provided.
3. The authors claim that the formation of ·NH radical is unlikely due to the inferior reaction kinetics as compared to ·NH₂. Can this be justified from the in situ EPR experiments?
4. The drop of CH₃CONH₂ yield is rather significant when the O₂ proportion increases from 75% to 100%. The EPR experiments suggest that this is due to the absence of ¹O₂ and the peroxidation induced by ·O₂·. Why does the O₂ increase in this O₂ proportion range could lead to a nearly total disappearance of ¹O₂? A more in-depth explanation of this result should be provided.
5. The reaction from CD₃-CDOD to CD₃CD₂O· is a tautomerization step rather than an oxidation step. The corresponding description for Figs 1 and 4 in the manuscript should be amended.

Reviewer #2

(Remarks to the Author)

In this work, Dong et al. propose a photocatalytic radical addition route to synthesize acetamide from mild co-oxidation of CH₃CH₂OH and NH₃. A comprehensive mechanism investigation (including iso-type (D and 15N) labeled in-situ EPR, HR-MS, NMR and in-situ ATR-FTIR) has been conducted to reveal the actual coupling intermediates and mechanism. The stable intermediates formation has been proved to be the decisive factor for achieving excellent catalytic activity. This work is meaningful for realizing the efficient production of amide compounds with a simple but efficient method for both industrial and fundamental studies. Overall, this work is very interesting, and the results can support most of their conclusions. I think it could be accepted after minor revision.

1. Line 202, “ynthrough” should be “through”.
2. In Figure S50 and 51, signal line for 0 min is missing in the G value around 3400 and 3420. Please check carefully of the full manuscript.

3. In the supplementary Figure 16, referring to Figure 15, the Y-axis information should be changed to the related compound concentration, in this case is NO₂⁻ concentration? The same change should also be made in Figure S17-S18, S20, S23.
4. In the control experiments of Figure 4e and Figure S43, the CH₃CONH₂ yield achieved in both experiments is nearly the same when using CH₃CH₂OH and CH₃CHO as the initial C reactant. It seems that the photooxidation of CH₃CH₂OH to CH₃CHO is very fast, so that the formation rate of CH₃CONH₂ is nearly the same when used CH₃CH₂OH and CH₃CHO as initial reactant. Please discuss and explain more about this phenomenon.
5. Why not directly use acetaldehyde to synthesize the C-N coupling products CH₃CONH₂ instead of ethanol. Actually, the alcohol oxidation to aldehyde will consume some system energy, and it seems that using aldehyde as initial reactant is a more energy saving choice. If change the initial concentration of CH₃CHO and NH₃, or change other reaction conditions, which authors have performed in the system of CH₃CH₂OH and NH₃, will the formation rate of CH₃CONH₂ can be improved further?
6. Please provide low resolution TEM images of TiO₂ before and after used in Figure S26. SEM images is hard to distinguish the particle size of TiO₂.
7. Authors stated that "The 1O₂ is generated from either the activation of the triplet oxygen (3O₂) or the h⁺-triggered rapid oxidation of superoxide radical". I am wondering to know that is there any experimental evidence can be used to support the formation of 3O₂ in this system? If the formation of 1O₂ is derived from the h⁺-triggered rapid oxidation of superoxide radical, but in the case of 100% concentration of O₂, there are no signals about 1O₂ detection. Usually, if more concentration of superoxide radical was formed, there would be more easily to convert 1O₂ according to the process authors proposed. Please explain more details about this phenomenon.
8. In this work, did the role of TiO₂ only absorb the photon and form the active electrons and holes in this system? Can other semiconductors also realize the same photocatalytic performance in this system? It seems that this reaction is not a semiconductor/catalysts sensitive.

Reviewer #3

(Remarks to the Author)

This work developed a heterogenous photocatalysis strategy for acetamide synthesis from the co-oxidation of NH₃ and EtOH reactants. A new reaction mechanism was proposed by the radical addition for tailoring the selectivity of C-N coupling. Direct experimental evidence, especially the comprehensive iso-type labeled in-situ characterizations, is provided to verify the precise reaction coordinates and critical intermediates, which presented record-high catalytic efficiencies for acetamide synthesis. This manuscript is well-organized with reliable results and delicate illustrations, which will have a significant impact on the research fields of catalysis and synthetic chemistry. Therefore, I recommend its acceptance in Nat. Commun. after addressing the following minor issues.

- 1) The superiority of this reaction route, in comparison with the conventional synthesis methods, should be indicated, including the synthesis efficiency, reaction mechanism and costs.
- 2) In a heterogeneous system, a decrease in mass transfer and light utilization efficiencies is inevitable. It should be illustrated how the high reaction rate is ensured at the light-irradiated catalysis interface.
- 3) More detailed reaction parameters for the route screening (Table 1) should be appended to guarantee its reproducibility.
- 4) A redox route of NH₃ oxidation and CH₃COOH reduction is conducted for route screening. It is confusing that only trace acetamide is fabricated since NH₃ and CH₃COOH are typical reactants for industrial acetamide synthesis. I assume the reason may be the proceeding neutral reaction in the liquid phase, which decreases the activity of NH₃ molecules and delivers CH₃COONH₄ as the side product. I recommend replacing CH₃COOH with CH₃COONa as the reductive reactant to test the catalytic efficiency.
- 5) Since the solid acetamide product is obtained, its NMR purity should be tested and appended in the revised manuscript.
- 6) The C- and N-based intermediates are accumulated for C-N coupling. Then how to exclude the possibility of C-C or N-N coupling?
- 7) The NH₂ species are detected by both FTIR and EPR measurements. What is the difference between the -NH₂ in FTIR and •NH₂ in EPR? How to distinguish these similar intermediates and how to identify their specific roles?
- 8) Only the effect of O₂ activation on the intermediate generation is illustrated by EPR. However, the key contribution of O₂ activation to the regulation of the reaction coordinates should be investigated, which may lead to different selectivity for the final products.
- 9) The structural formula of CH₃•CHOH (Fig. S37) is not consistent with those in the main manuscript. Also, I believe the trapping agent should be TEMPO rather than TEMP.

Reviewer #4

(Remarks to the Author)

In this manuscript, the authors present a method for synthesizing acetamide from various nitrogen and carbon sources. I appreciate the extensive number of experiments conducted in this study, and the authors claim very high selectivity and activity of the product. However, there are many unclear points regarding the results and methodology, making it unsuitable for publication in Nature Communications. Please see my comments below.

1. The intensity (y-value) in the y-axis of the calibration experiments/curves in Figure S1 should be provided and started from 0. Why the b value (83840.0 a.u?) in the equation of the standard curve is very large? Does it mean that at zero formamide concentration, the signals of this substance in the HPLC are already very substantially high? Why is that? Similar phenomena can be seen in the Figure S2 and S3. How the intensity in the y-axis in these standard curves was measured? It is the height of the peak or the area of the peak? A similar query is raised for Figure S9.

2. Why does double catalyst dosage (Figure 2a), from 1 to 2 mg, result in a nearly 8 times increase in production rate? As the Authors proved that P25 does not change throughout the durability test, so by doubling catalyst loading, we can assume that the active surface area is doubled, leading to doubled amounts of products.
3. In Figure 2b, the legend shows 8 different N-sources, but the data only shows 4 lines. A similar question is asked in Figure 2c. Is it because some of the results are nearly zero?
4. In Supplementary Note 5, the Authors mentioned that the reaction solution was extracted for the CH₃CONH₂ every 10 h for the stability test, but why the data points in Figure 2e do not show at every 10 h?
5. From the long-term stability test (Figure 2e), we can see that the amount of products linearly increases over time, reaching 1.82 g of acetamide after 300 h, which is equivalent to ~0.03 mole of NH₃. However, the total mole of NH₃ added to the reactor is only ~0.0024 moles (800 mg L⁻¹ NH₃ in CH₃CH₂OH, and only 50 mL of CH₃CH₂OH was used, so the amount of NH₃ is 40 mg). Why it is possible?
6. From the in situ ATR-FTIR in Figure 3c,f,g, we can see that the amount of C-N coupling is saturated after around 20 min of irradiation, so why in the long-term test the total production keeps increasing linearly over time?
7. In Figure 3a, what are the peaks for the spectrum in the dark? Why the C-OH peak is not observed in for this sample in the dark? Similarly, why there is no peak (N-H) for ammonia in Figure 3b in the dark? In Figure 3c, after mixing CH₃CH₂OH and NH₃, why the peaks appear in Figure 2a for OER does not show in this Figure in the dark?
8. It is interesting when the data in Figure 3a-c in the dark and after 10 min of irradiation are substantially different. If the reactions occur, should the products/intermediates appear gradually rather than substantially change? So we should expect a gradual change between the dark and 10 min irradiation. Can it be revealed if the spectra for the shorter irradiation time are obtained?
9. Another critical point is that the authors compare their work with different studies in the literature (Fig 2d and Table S1). However, there are too many differences between the methodologies used to collect data, so I don't believe these comparisons are accurate. For example, when electrocatalysis is used, production rates can depend on applied currents, potentials, electrolytes, and cell configurations. It is also difficult to justify comparisons between photocatalysis and electrocatalysis. In photocatalysis, light intensity is the most important factor, and I don't think the light intensity is reported in this work. If the light intensity differs from that in other studies, then there is no valid basis for comparison.
10. A typo in Supplementary Note 4. The last sentence should be C-source, not N-sources.

Version 1:

Reviewer comments:

Reviewer #1

(Remarks to the Author)

All my comments have been addressed, and this manuscript is publishable in the present form.

Reviewer #2

(Remarks to the Author)

I am satisfied with those modifications, I recommend accept this manuscript with further revision.

Reviewer #3

(Remarks to the Author)

The authors reported a new photocatalytic radical addition route for gram-scale acetamide synthesis, which is interesting. Meanwhile, the authors revised the manuscript well and addressed most of the questions. Therefore, I would suggest acceptance in Nat Commun.

Reviewer #4

(Remarks to the Author)

Thank you to the authors for their revision. However, given the substantial discrepancies between the data in the initial submission and the revised version, which changes the nature of the methodology used in this work and the reported results, I do not believe this manuscript meets the standards for publication in Nature Communications.

For my comment 2, the catalyst loading for Figure 2a has been updated to 10 times higher than in the initial submission. The authors attributed this to a 'graphical error.' However, such an error could have led to highly misleading results, making it difficult to compare with future papers. More critically, I find the explanation of a 'graphical error' oversimplified, especially given that the authors simply increased the catalyst loading in response. This raises concerns about the accuracy of the reported catalyst amounts. Even with the revised data, increasing the catalyst loading by 5x resulted in a performance improvement of more than 7.5x, which ties back to my previous question regarding the performance increase.

For my comment 5, the authors have now added the following sentence to the Supporting Information: 'The NH₃ is periodically provided. When the NH₃ is depleted, 800.00 mg L⁻¹ of NH₃ is added to the long-term stability test system.' However, I question why this critical detail was omitted from the initial submission. Without the periodic addition of NH₃, the results of this study could have been drastically different, potentially misleading readers upon publication. Even after adding this statement, it remains unclear when NH₃ was added. The authors only mention it was added when 'depleted,' but when

exactly was that? How many times was NH₃ added, and what was the total amount added? This would directly affect the final production yield and should be clarified.

I am unclear on the authors' response to my comment 6. As I initially mentioned, when the FTIR results show no further change, it suggests that either the product concentration has reached saturation or it exceeds the detection limit of the equipment. However, the response provided by the authors seems unclear and does not fully address this point. Additionally, I have another question regarding the in situ measurements: in Figures 3e, f, and g, the Y-axis is labeled as 'Normalized abs.' What exactly is the data normalized to? Is it normalized to the highest data point, the catalyst loading, or something else? Clarifying this is essential for interpreting the presented data."

For my comments 7 and 8, if you overlay the spectra at 2, 4, 6, 8, and 10 minutes in Figures S37, S38, and S39, there appears to be no significant difference between them. This raises the question of why there is a substantial change from 0 to 2 minutes, but almost no change after 10 minutes, with only a slight variation in peak intensity after 20, 30, and 40 minutes.

For my comment 9, I do not feel that the authors have adequately addressed my question. While the revised version now includes details on the light sources used in both this work and the cited literature, the light intensity is still not reported. Using the same light source, such as a 300 W Xe lamp, can lead to significantly different outcomes depending on the intensity applied. For example, one experiment might use 1 sun (100 mW cm⁻²), while another could use 2 sun or even 7 sun, which can substantially (and non-linearly) alter the production yield. This critical information must be provided to allow for accurate comparisons and reproducibility.

Response to the Reviewers' Comments

Response to Referee 1:

General Comments: The manuscript authored by Li et al demonstrates that a couple of stable-transient intermediates generated during photocatalytic reactions can offer a high opportunity for C-N coupling. In their work, CH₃CH₂OH is used as the C-source, which undergoes oxidation into a stable intermediate, CH₃CHO; NH₃ is the N-source, the amount of which is limited so as to control the generation of a transient intermediate, ·NH₂. The fast coupling reaction between these two intermediates leads to a high CH₃CONH₂ synthesis rate and a promising reaction selectivity. This reaction route is innovative, and the in situ measurements have provided reliable evidence for both intermediates participating in the reaction pathways. The manuscript is publishable provided the following comments are addressed:

Response: Thanks for the comments and helpful suggestions. We have addressed the concerns in the following point-by-point response.

Comment 1: The discussion on the role of photocatalyst is missing in this manuscript. How does TiO₂ steer the oxidative reactions of both CH₃CH₂OH and NH₃? It would be best if the authors could provide some test for other photocatalysts and justify that the performances are similar. Otherwise, the catalyst characteristics (catalytic active sites) may also be a decisive factor.

Response: Thanks for the comments. The role of TiO₂ in this manuscript is to provide photoinduced e⁻ and h⁺, which is a fundamental application of general photocatalysts. The h⁺ is used to activate NH₃ and CH₃CH₂OH and the e⁻ is used to activate O₂, driving the oxidation selectivity. Regarding the specificity of photocatalysts, we screened some common photocatalysts according to your suggestion. Among the selected catalysts, TiO₂ has been confirmed to show the best photocatalytic activity due to its valence and conduction bands' moderate position, leading to mild oxidizing and reducing ability. The other catalysts also have photocatalytic C-N coupling activity with similar performance. However, it is undeniable that the photocatalytic C-N coupling activity can be enhanced if some active sites can be constructed on TiO₂. The discussion on the

role of photocatalyst is added in the Manuscript and the experiment details for screening common photocatalysts in Supplementary Information are as follows:

Page 5 in Manuscript: The commercial TiO₂ (P25) is applied as a representative photocatalyst to develop the photocatalytic redox route for C-N coupling, which only absorbs the photon and forms the active electrons and holes in this system, excluding the specificity of other modified photocatalysts and establishes a general pattern for the superiority of the photocatalytic amides synthesis scheme. Other common photocatalysts (BiOCl, BiOBr, Bi₂O₂CO₃, Sr₂Sb₂O₇, and g-C₃N₄) are also investigated in the C-N coupling reaction, which processes similar properties, showing the universality of photocatalysts in C-N coupling (Supplementary Figure 9 and Supplementary Note 3).

Supplementary Note 3 | Photocatalyst screening experiment.

1) The preparation of photocatalyst:

Commercial TiO₂ was purchased from Degussa Co. Ltd.

Anatase TiO₂ was purchased from Macklin Co. Ltd.

Nanosheets TiO₂ were prepared by the hydrothermal method by mixing 20.00 mL of tetrabutyl titanate and 3.00 mL of hydrofluoric acid solution in a dry Teflon-lined stainless-steel autoclave of 100.00 mL capacity and then stored in an electric furnace at 180 °C for 24h. After the hydrothermal reaction, the autoclave was cooled naturally to room temperature, and the white solid was separated by high-speed centrifugation. It was washed several times with ethanol and deionized water, dried at 80 °C for 12 h, and labeled as the TNS.

BiOCl: 4.00 mmol Bi(NO₃)·5H₂O and 4.00 mmol KCl were dissolved in 100.00 mL deionized water, followed by vigorous stirring at room temperature for one hour. After the reaction, the precipitates were collected by centrifugation, washed with deionized water and ethanol, and dried at 60 °C in a vacuum. The obtained products were labeled as BiOCl.

BiOBr: 4.00 mmol Bi(NO₃)·5H₂O and 4.00 mmol KBr were dissolved in 100.00 mL deionized water, followed by vigorous stirring at room temperature for one hour. The rest of the operation is consistent with BiOCl. The obtained products were labeled as

BiOBr.

Bi₂O₂CO₃: 2.00 mmol Bi(NO₃)·5H₂O were dissolved in 100.00 mL deionized water, and stirred at room temperature for one hour. Subsequently, 6.00 mmol urea was added to the solution and stirred for 30 min. The precipitates were separated by centrifugation and washed with deionized water and absolute ethanol three times. The obtained solids were collected and dried overnight at 180 °C, being further labeled as Bi₂O₂CO₃.

Sr₂Sb₂O₇: 5.00 mmol C₄H₆O₄Sr and 2.50 mmol K₂H₂Sb₂O₇ were dissolved in 30.00 mL of deionized water. Under vigorous stirring, HNO₃ or NaOH was added dropwise into the solution until pH = 13. Then, the solution was moved into a 45.00 mL Teflon-lined autoclave and heated at 180 °C for 48 h. After cooling in the air, the obtained mixture was centrifuged several times with deionized water and ethanol successively. Finally, the collected samples were dried in an oven at 60 °C for a whole night. The resulting product was labeled as Sr₂Sb₂O₇.

g-C₃N₄: 10.00 g of urea were placed in an alumina crucible with a cover. The crucible was heated to 550 °C at a heating rate of 15 °C·min⁻¹ in a muffle furnace and maintained for 2 h. After the thermal treatment, the crucible was cooled down to room temperature in the muffle furnace. The samples collected were labeled as g-C₃N₄.

2) Photocatalyst screening experiment.

The total reaction volume was set to 50.00 mL, which consists of 49.00 mL of CH₃CN as the solvent. 1.00 mL of CH₃CH₂OH and 800.00 mg L⁻¹ of NH₃ (in CH₃CH₂OH) are introduced as the C- and N-sources respectively. 75% of O₂ (in Ar) was continuously injected. 5.00 mg of different photocatalysts (such as commercial TiO₂, nanosheet TiO₂, anatase TiO₂, BiOCl, BiOBr, Bi₂O₂CO₃, Sr₂Sb₂O₇, and g-C₃N₄) were dispersed in the reaction solution respectively. The other experimental details are the same as that of the Supplementary Note 1.

Supplementary Figure 9 | C-N coupling performance of different photocatalysts.

Comment 2: Figs 3b and c indicate a markedly increased amount of NH_2OH species during the reactions, but it has not been displayed in Figs 2 and 4. A detailed evaluation of the role of NH_2OH should be provided.

Response: Thanks for the comments. Regarding the issues of NH_2OH species, we introduced *in-situ* UV experiments for detecting NH_2OH species when evaluating N-selectivity and investigating C-N coupling intermediates. However, there is no absorbance in 709 nm in Supplementary Figure 20, which is referred to as NH_2OH . The experiment details are demonstrated in Supplementary Note 14. According to Figures 3b and 3c in the manuscript, the NH_2OH species are mostly adsorbed on the surface of the catalyst, as an intermediate product of AOR. However, as the AOR proceeds, NH_2OH is further oxidized, upon forming and adsorbing on the surface, it diffuses to the solution and combines with other intermediates rapidly, which is transferred to the C-N coupling production. Due to the extremely rapid oxidation of ammonia and the inherent instability of the NH_2OH species, the accumulation and detection of NH_2OH are impeded. The corresponding results and discussion are included in the revised files as follows:

Page 9 in Manuscript: Since NH_2OH species are not detected during the reaction process (Figure 2b, Supplementary Figure 20), it is clarified that NH_2OH is not desorbed from the catalyst surface but is instead converted to other intermediates and products.

Supplementary Note 14 | Experimental details for NH₂OH detection through colorimetric method.

Firstly, 1.00 mL of the reaction solution with the photocatalyst filtered out was diluted with 1.00 mL of 0.05 M phosphate buffer solution (pH=6.8) and 0.80 mL of deionized water. Then, 0.20 mL of trichloroacetic acid, 1.00 mL of 1% 8-quinolinol (w/v ethanol), and 1.00 mL of 1 M Na₂CO₃ solution were injected into the above solution. Finally, the solution was heated in a water bath under the dark at 100 °C for 1 min, and then the absorbance at 709 nm was measured using a UV-visible absorption spectrophotometer.

Supplementary Figure 20 | Parallel experiments for the NH₂OH (a and b) for the CH₃CONH₂ selectivity (N) test.

Supplementary Figure 35 | Tested standard intensities (a) and curves (b) for NH₂OH detection by a UV-vis.

Comment 3: The authors claim that the formation of ·NH radical is unlikely due to the inferior reaction kinetics as compared to ·NH₂. Can this be justified from the in

situ EPR experiments?

Response: Thanks for the comments. According to your suggestion, we made another effort to detect $\bullet\text{NH}$ radicals. However, by using the current *in-situ* EPR technology, we can only detect $\bullet\text{NH}_2$ radical, as exhibited below. There is a time gap between continuous sampling points, causing the short-lifetime radicals not to be detected through *in-situ* EPR. The lifetime of $\bullet\text{NH}_2$ radical and $\bullet\text{NH}$ radical are 1 ns and 13 ps respectively, according to the Reference: Fuke, K. & Takasu, R. Ultrafast Photochemistry of Ammonia Clusters: Formation and Decay of Hypervalent Molecular Clusters Containing the NH_4 Radical. *Bull. Chem. Soc. Japan* **68**, 3309-3318 (2006); Savee, J. D., Mann, J. E. & Continetti, R. E. Stability of the Ground and Low-Lying Vibrational States of the Ammonium Radical. *J. Phys. Chem. Lett.* **4**, 3683-3686 (2013). Additionally, the selectivity of DMPO is another important factor in radical trapping experiment, which also impedes the detection of $\bullet\text{NH}$ radicals.

Supplementary Figure 56 | *In-situ* EPR results for the detection of DMPO- $\bullet^{14}\text{NH}_2$ (a) and DMPO- $\bullet^{15}\text{NH}_2$ (b) respectively.

Comment 4: The drop of CH_3CONH_2 yield is rather significant when the O_2 proportion increases from 75% to 100%. The EPR experiments suggest that this is due to the absence of $^1\text{O}_2$ and the peroxidation induced by $\bullet\text{O}_2$. Why does the O_2 increase in this O_2 proportion range could lead to a nearly total disappearance of $^1\text{O}_2$? A more in-depth explanation of this result should be provided.

Response: Thanks for the comments. We agree that as O_2 concentration increases the

$^1\text{O}_2$ signal gradually disappears and the $\bullet\text{O}_2^-$ radical signal gradually increases. For this reason, we have done more subdivided *in-situ* EPR experiments at O_2 concentration to support this conclusion. At the O_2 concentration of 90%, $\bullet\text{O}_2^-$ radical signal is weaker than that at 100% concentration, and the $^1\text{O}_2$ signal at 90% O_2 concentration is weaker than that at 75% concentration, which suggests that there is a gradual evolution of the $^1\text{O}_2$ to $\bullet\text{O}_2^-$ in the range of 75%-100% O_2 concentration. More details are added in Supplementary Figure 50.

Page 12 in Manuscript: As the concentration of O_2 increases, the signal intensity of $\bullet\text{O}_2^-$ radical gradually increases while the signal intensity of $^1\text{O}_2$ gradually decreases. Finally, the excess provision of O_2 (100%) leads to the accumulation of the strong oxidative $\bullet\text{O}_2^-$ radicals (Figure 4d, Supplementary Figure 48 and Supplementary Note 20), in which no signals of the $^1\text{O}_2$ are detected (Figure 4d and Supplementary Figures 49 and 50). As the $\bullet\text{O}_2^-$ radicals bind to the reactants to generate peroxidation products, $^1\text{O}_2$ species are no longer generated.

Supplementary Figure 50 | *In-situ* EPR results for (a) TEMP-trapping and (b) DMPO-trapping under the O_2 proportion of 75%, 90% and 100% (d) respectively.

Comment 5: *The reaction from $\text{CD}_3\text{-CDOD}$ to $\text{CD}_3\text{CD}_2\text{O}\cdot$ is a tautomerization step rather than an oxidation step. The corresponding description for Figs 1 and 4 in the manuscript should be amended.*

Response: Thanks for the suggestions. This confusing description is now revised in the Manuscript (Figures 1 and 4).

Page 12 in Manuscript: There is a tautomerization transformation from $\text{CD}_3\bullet\text{CDOD}$ to $\text{CD}_3\text{CD}_2\text{O}\bullet$, which is driven by the mild oxidative ability of singlet oxygen ($^1\text{O}_2$, Figure 4c and Supplementary Figure 47).

Page 14 in Manuscript: The provision of $\text{CH}_3\text{CH}_2\text{OH}/\text{CD}_3\text{CD}_2\text{OD}$ feedstock results in the generation of H/D labeled acetaldehyde and acetamide respectively, presenting a selective EOR along the route of $\text{CH}_3\text{CH}_2\text{OH} \rightarrow \text{CH}_3\bullet\text{CHOH} \rightarrow \text{CH}_3\text{CH}_2\text{O}\bullet \rightarrow \text{CH}_3\text{CHO} \rightarrow \text{CH}_3\text{CONH}_2$.

Figure 1. Illustration for the coupling of stable and transient intermediates for acetamide photosynthesis.

Figure 4. Generation pathways and mechanism for the coupling intermediates: (a) O_2 proportion-dependent CH_3CONH_2 yield; DMPO-trapping (with D-labeling, images above) and TEMP-trapping (images below) *in-situ* EPR experiments under the O_2 proportion of 0% (b), 75% (c) and 100% (d) respectively; (e) Control experiments for CH_3CONH_2 synthesis by replacing CH_3CH_2OH with CH_3CHO as the C-source; GC (f) and 1H NMR (g) results with D- (images right) and H-labeling (images left) respectively; (h) DMPO-trapping *in-situ* EPR experiments with ^{14}N - (image above) and ^{15}N -labeling (image below) respectively; (i) *in-situ* EPR spectra for e^- -induced TEMPO consumption with (image below) and without (image above) NH_3 provision respectively; (j) HR-MS results for CH_3CONH_2 detection with ^{14}N - (image above) and ^{15}N -labeling (image below) respectively; (k) Proposed reaction pathways of CH_3CH_2OH and NH_3 co-oxidation for intermediates coupling and CH_3CONH_2 synthesis. The error bars in (a) and (e) were drawn based on the calculated standard error of two parallel tests.

Response to Referee 2:

General Comments: In this work, Dong et al. propose a photocatalytic radical addition route to synthesize acetamide from mild co-oxidation of CH₃CH₂OH and NH₃. A comprehensive mechanism investigation (including iso-type (D and ¹⁵N) labeled in-situ EPR, HR-MS, NMR and in-situ ATR-FTIR) has been conducted to reveal the actual coupling intermediates and mechanism. The stable intermediates formation has been proved to be the decisive factor for achieving excellent catalytic activity. This work is meaningful for realizing the efficient production of amide compounds with a simple but efficient method for both industrial and fundamental studies. Overall, this work is very interesting, and the results can support most of their conclusions. I think it could be accepted after minor revision.

Response: Thanks for the comments and helpful suggestions. We have addressed the concerns in the following point-by-point response.

Comment 1: Line 202, “ynthrogh” should be “through”.

Response: Thanks for the comments. We have corrected this mistake in the manuscript.

Page 10 in Manuscript: Therefore, it is identified that the amide bond (-CONH₂) can be effectively **synthesized through** the coupling of C=O and -NH₂-based intermediates.

Comment 2: In Figure S50 and 51, signal line for 0 min is missing in the G value around 3400 and 3420. Please check carefully of the full manuscript.

Response: Thanks for the comments. Please allow us to offer a more detailed explanation for Figures S50 and S51, which are now labeled as S58 and S59 respectively. TEMPO itself is paramagnetic and presents characteristic signals of three peaks in EPR spectra, with the same initial concentration and the same initial intensity of the TEMPO signal. In a photocatalytic reaction, electrons combine with TEMPO to form TEMPOH, causing a decrease in signal intensity. Hence, it is illustrated that the effect of NH₃ introduction is on the decreasing rate of TEMPO, in which the signal lines of 0 min are the curves with the highest intensity (Supplementary Figures 58 and 59).

Supplementary Figure 58 | *In-situ* EPR results for the detection of TEMPO reduction by light-generated e^- without NH_3 oxidation cooperation.

Supplementary Figure 59 | *In-situ* EPR results for the detection of TEMPO reduction by light-generated e^- with NH_3 oxidation cooperation.

Comment 3: *In the supplementary Figure 16, referring to Figure 15, the Y-axial information should be changed to the related compound concentration, in this case is NO_2^- concentration? The same change should also be made in Figure S17-S18, S20, S23.*

Response: Thanks for the comments. The reason for the misunderstanding of the Y-axial information is that we have adopted different detection methods for the gas phase

and the liquid phase products. Supplementary Figures 11, 16-18, 20, and 23 were detected by HPLC and IC for identification of products at the liquid phase, whose Y-axis is the intensity of the target peaks. However, Supplementary Figures 12-15, 19, 21-22 are detected by an infrared gas analyzer for the identification of products at the gas phase. The analyzer performs integration by the Infrared flue gas analyzer. The analyzer integrates the infrared spectra of the resolved gas products and obtains the corresponding concentrations, based on the instrument's internal procedures and calibration curves. Therefore, the Y-axis represents the concentration of the target product. To avoid further misunderstanding, we have added a note (Supplementary Note 15) to explain the reasonableness of multiple methods for detecting C/N species.

Supplementary Note 15 | The reason for applying different methods for detecting C/N species.

For the Efficiency evaluation for acetamide photosynthesis (unit production rate, C/N selectivity), we chose to evaluate the activity of C-N coupling: using the combination of multiple detection techniques:

Gas-phase products: N species including N_2 , N_2O , NO , NO_2 and C species including CO_2 , CO and CH_3CHO are detected by an infrared flue gas analyzer.

Liquid phase products: NH_4^+ by cation chromatography, NO_3^- , NO_2^- and CH_3COO^- by anion chromatography and the target product CH_3CONH_2 by HPLC.

Through the above methods, we can comprehensively test all possible C/N products and achieve scientific performance evaluation.

Comment 4: In the control experiments of Figure 4e and Figure S43, the CH_3CONH_2 yield achieved in both experiments is nearly the same when using CH_3CH_2OH and CH_3CHO as the initial C reactant. It seems that the photooxidation of CH_3CH_2OH to CH_3CHO is very fast, so that the formation rate of CH_3CONH_2 is nearly the same when used CH_3CH_2OH and CH_3CHO as initial reactant. Please discuss and explain more about this phenomenon.

Response: Thanks for the comments. By combining *in-situ* experiments and

photocatalyst performance experiments, we found that $\text{CH}_3\text{CH}_2\text{OH}$ oxidizes rapidly to obtain CH_3CHO , and the CH_3CHO will not accumulate but participates in C-N coupling (Figures 2c and 2f). When we use CH_3CHO instead of $\text{CH}_3\text{CH}_2\text{OH}$ as the initial reactant, the nearly identical CH_3CONH_2 yield confirms CH_3CHO as a key intermediate for C-N coupling (Figure 4e). In the reaction process, we should precisely regulate the oxidation of $\text{CH}_3\text{CH}_2\text{OH}$ to avoid excessive accumulation of CH_3CHO , which decreases the selectivity of acetamide. The respective description has been added to the Manuscript.

Page 14 in Manuscript: It is noted that almost complete recovery of the CH_3CONH_2 yield is accomplished via the NH_3 - CH_3CHO condensation ($204.67 \pm 6.58 \mu\text{mol h}^{-1}$), consistent with that of the co-oxidation of NH_3 and $\text{CH}_3\text{CH}_2\text{OH}$ ($209.77 \pm 7.68 \mu\text{mol h}^{-1}$), which identifies CH_3CHO as the stable C-intermediate, in which the generation of CH_3CHO requires precise regulation to avoid the decrease of the acetamide.

Comment 5: Why not directly use acetaldehyde to synthesize the C-N coupling products CH_3CONH_2 instead of ethanol. Actually, the alcohol oxidation to aldehyde will consume some system energy, and it seems that using aldehyde as initial reactant is a more energy saving choice. If change the initial concentration of CH_3CHO and NH_3 , or change other reaction conditions, which authors have performed in the system of $\text{CH}_3\text{CH}_2\text{OH}$ and NH_3 , will the formation rate of CH_3CONH_2 can be improved further.

Response: Thanks for the comments. As to why CH_3CHO is not chosen to synthesize CH_3CONH_2 , it is illustrated from the point of view of price, purity and stability. First, CH_3CHO in industrial production is 1430 USD/MT, which is also unstable with low purity, volatile, and possessing certain toxicity. However, the price of $\text{CH}_3\text{CH}_2\text{OH}$ in industrial production is 586 USD/MT, and absolute $\text{CH}_3\text{CH}_2\text{OH}$ has high purity and good stability. Therefore, $\text{CH}_3\text{CH}_2\text{OH}$ is more suitable for C-N coupling than CH_3CHO . After selecting $\text{CH}_3\text{CH}_2\text{OH}$ as the reactant, we carried out a series of condition optimization (Supplementary Figures 4-9), comprehensively considered the yield and selectivity of CH_3CONH_2 , and finally selected the C-N coupling from $\text{CH}_3\text{CH}_2\text{OH}$ and

NH₃.

Page 13 in Manuscript: The corresponding control experiment is thereby conducted by replacing CH₃CH₂OH with CH₃CHO as the initial reactant for C-N coupling (Figure 4e, Supplementary Figure 51 and Supplementary Table 6).

Supplementary Table 6 | Comparison of CH₃CH₂OH and CH₃CHO as the reactants.

	Price, USD/MT	Purity	stability
CH ₃ CHO	1430	40% aqueous solution of acetaldehyde	unstable
CH ₃ CH ₂ OH	586	99.5% ethanol absolute	stable

Comment 6: Please provide low resolution TEM images of TiO₂ before and after used in Figure S26. SEM images is hard to distinguish the particle size of TiO₂.

Response: Thanks for the comment. The low-resolution TEM images of TiO₂ before and after application in long-term stability tests have been added in Supplementary Figure 28.

Page 8 in Manuscript: No deactivation of the P25 is observed based on the characterization results of the sample before and after the long-term stability test (Supplementary Figures. 27-30 and Supplementary Note 18).

Supplementary Figure 29 | Low-resolution TEM images for P25 before (a) and after (b) the long-term stability test.

Comment 7: Authors stated that “The ¹O₂ is generated from either the activation of

the triplet oxygen ($^3\text{O}_2$) or the h^+ -triggered rapid oxidation of superoxide radical". I am wondering to know that is there any experimental evidence can be used to support the formation of $^3\text{O}_2$ in this system? If the formation of $^1\text{O}_2$ is derived from the h^+ -triggered rapid oxidation of superoxide radical, but in the case of 100% concentration of O_2 , there are no signals about $^1\text{O}_2$ detection. Usually, if more concentration of superoxide radical was formed, there would be more easily to convert $^1\text{O}_2$ according to the process authors proposed. Please explain more details about this phenomenon.

Response: Thanks for the comments. Regarding the issue of the $^1\text{O}_2$ generation, a more detailed explanation are provided below. There are two main pathways to generate $^1\text{O}_2$, one is photocatalyst transfers energy to the $^3\text{O}_2$, which is the ground state of O_2 ; The other is the oxidation of $\bullet\text{O}_2^-$ by h^+ . Since we injected O_2 to simulate the reaction process in the *in-situ* EPR experiment, we cannot rule out the presence of $^3\text{O}_2$. However, the available technology cannot tell which one is the real formation of the pathway, all we can do is hypothesize based on the literature and data. As for your second question, if more concentration of $\bullet\text{O}_2^-$ was formed, it would be easier to convert $^1\text{O}_2$, we believe that when the concentration of $\bullet\text{O}_2^-$ is too high, $\bullet\text{O}_2^-$ will preferentially react with NH_3 and $\text{CH}_3\text{CH}_2\text{OH}$ to produce peroxide products (as shown in Supplementary Figures 42 and 50, $\text{CH}_3\text{CH}_2\text{OH}$ yield decreased while CH_3COOH and $\text{NO}_3^-/\text{NO}_2^-$ yield increased at 100% O_2 concentration), instead of transforming into $^1\text{O}_2$. Therefore, in the case of 100% concentration of O_2 , there are no signals about $^1\text{O}_2$ detection.

Page 12 in Manuscript: The $^1\text{O}_2$ is generated from either the activation of the triplet oxygen ($^3\text{O}_2$) or the h^+ -triggered rapid oxidation of superoxide radical ($\bullet\text{O}_2^-$).^{44,45} Since $^3\text{O}_2$ and $\bullet\text{O}_2^-$ are difficult to distinguish in the current *in-situ* EPR experiments, it is reasonable to infer that both of these two reaction pathways are involved in the generation of $^1\text{O}_2$.

Page 12 in Manuscript:

As the concentration of O_2 increases, the signal intensity of $\bullet\text{O}_2^-$ radical gradually increases while the signal intensity of $^1\text{O}_2$ gradually decreases. Finally, the excess provision of O_2 (100%) leads to the accumulation of the strong oxidative $\bullet\text{O}_2^-$ radicals

(Figure 4d, Supplementary Figure 48 and Supplementary Note 20), in which no signals of the $^1\text{O}_2$ are detected (Figure 4d, Supplementary Figures 49 and 50). As the $\bullet\text{O}_2^-$ radicals bind to the reactants to generate peroxidation products, $^1\text{O}_2$ species are no longer generated.

Supplementary Figure 41 | Products concentration of photocatalytic C-N coupling at different oxygen concentrations. (a) and (b) under the O₂ proportion of 0%; (c) and (d) under the O₂ proportion of 75%; (e) and (f) under the O₂ proportion of 100%.

Supplementary Figure 49 | *In-situ* EPR results for the detection of TEMPO from the generation of TEMP oxidation by ¹O₂ at the O₂ proportion of 100%.

Supplementary Figure 50 | *In-situ* EPR results for (a) TEMP-trapping and (b) DMPO-trapping under the O₂ proportion of 75%, 90% and 100% (d) respectively.

Comment 8: *In this work, did the role of TiO₂ only absorb the photon and form the active electrons and holes in this system? Can other semiconductors also realize the same photocatalytic performance in this system? It seems that this reaction is not a semiconductor/catalysts sensitive.*

Response: Thanks for the comments. Regarding the role of TiO₂ in the C-N coupling reaction, it is clarified that no specific active sites are provided, in which the active e⁻

h^+ pairs are produced at their suitable band edges. For this reason, we supplemented the screening experiments with common photocatalysts and found that the majority of photocatalysts have C-N coupling activity, proving this reaction is not semiconductor/catalysts sensitive. However, TiO_2 has the best activity due to its more suitable valence band position and moderate redox capacity.

Page 5 in Manuscript: The commercial TiO_2 (P25) is applied as a representative photocatalyst to develop the photocatalytic redox route for C-N coupling, which only absorbs the photon and forms the active electrons and holes in this system, excluding the specificity of other modified photocatalysts and establishes a general pattern for the superiority of the photocatalytic amides synthesis scheme. Other common photocatalysts are also investigated in the C-N coupling reaction, which processes similar properties, showing the universality of photocatalysts in C-N coupling (Supplementary Figure 4 and Supplementary Note 3).

Supplementary Note 3 | Photocatalyst screening experiment.

1) The preparation of photocatalyst:

Commercial TiO_2 was purchased from Degussa Co. Ltd.

Anatase TiO_2 was purchased from Macklin Co. Ltd.

Nanosheets TiO_2 were prepared by the hydrothermal method by mixing 20.00 mL of tetrabutyl titanate and 3.00 mL of hydrofluoric acid solution in a dry Teflon-lined stainless-steel autoclave of 100.00 mL capacity and then stored in an electric furnace at 180 °C for 24h. After the hydrothermal reaction, the autoclave was cooled naturally to room temperature, and the white solid was separated by high-speed centrifugation. It was washed several times with ethanol and deionized water, dried at 80 °C for 12 h, and labeled as the TNS.

$BiOCl$: 4.00 mmol $Bi(NO_3)_3 \cdot 5H_2O$ and 4.00 mmol KCl were dissolved in 100.00 mL deionized water, followed by vigorous stirring at room temperature for one hour. After the reaction, the precipitates were collected by centrifugation, washed with deionized water and ethanol, and dried at 60 °C in a vacuum. The obtained products were labeled as $BiOCl$.

$BiOBr$: 4.00 mmol $Bi(NO_3)_3 \cdot 5H_2O$ and 4.00 mmol KBr were dissolved in 100.00 mL

deionized water, followed by vigorous stirring at room temperature for one hour. The rest of the operation is consistent with BiOCl. The obtained products were labeled as BiOBr.

$\text{Bi}_2\text{O}_2\text{CO}_3$: 2.00 mmol $\text{Bi}(\text{NO}_3)_3 \cdot 5\text{H}_2\text{O}$ were dissolved in 100.00 mL deionized water, and stirred at room temperature for one hour. Subsequently, 6.00 mmol urea was added to the solution and stirred for 30 min. The precipitates were separated by centrifugation and washed with deionized water and absolute ethanol three times. The obtained solids were collected and dried overnight at 180 °C, being further labeled as $\text{Bi}_2\text{O}_2\text{CO}_3$.

$\text{Sr}_2\text{Sb}_2\text{O}_7$: 5.00 mmol $\text{C}_4\text{H}_6\text{O}_4\text{Sr}$ and 2.50 mmol $\text{K}_2\text{H}_2\text{Sb}_2\text{O}_7$ were dissolved in 30.00 mL of deionized water. Under vigorous stirring, HNO_3 or NaOH was added dropwise into the solution until $\text{pH} = 13$. Then, the solution was moved into a 45.00 mL Teflon-lined autoclave and heated at 180 °C for 48 h. After cooling in the air, the obtained mixture was centrifuged several times with deionized water and ethanol successively. Finally, the collected samples were dried in an oven at 60 °C for a whole night. The resulting product was labeled as $\text{Sr}_2\text{Sb}_2\text{O}_7$.

g- C_3N_4 : 10.00 g of urea were placed in an alumina crucible with a cover. The crucible was heated to 550 °C at a heating rate of $15\text{ }^\circ\text{C} \cdot \text{min}^{-1}$ in a muffle furnace and maintained for 2 h. After the thermal treatment, the crucible was cooled down to room temperature in the muffle furnace. The samples collected were labeled as g- C_3N_4 .

2) Photocatalyst screening experiment.

The total reaction volume was set to 50.00 mL, which consists of 49.00 mL of CH_3CN as the solvent. 1.00 mL of $\text{CH}_3\text{CH}_2\text{OH}$ and 800.00 mg L^{-1} of NH_3 (in $\text{CH}_3\text{CH}_2\text{OH}$) are introduced as the C- and N-sources respectively. 75% of O_2 (in Ar) was continuously injected. 5.00 mg of different photocatalysts (such as commercial TiO_2 , nanosheet TiO_2 , anatase TiO_2 , BiOCl, BiOBr, $\text{Bi}_2\text{O}_2\text{CO}_3$, $\text{Sr}_2\text{Sb}_2\text{O}_7$, and g- C_3N_4) were dispersed in the reaction solution respectively. The other experimental details are the same as that of the Supplementary Note 1.

Supplementary Figure 4 | C-N coupling performance of different photocatalysts.

Response to Referee 3:

General Comments: This work developed a heterogenous photocatalysis strategy for acetamide synthesis from the co-oxidation of NH₃ and EtOH reactants. A new reaction mechanism was proposed by the radical addition for tailoring the selectivity of C-N coupling. Direct experimental evidence, especially the comprehensive iso-type labeled in-situ characterizations, is provided to verify the precise reaction coordinates and critical intermediates, which presented record-high catalytic efficiencies for acetamide synthesis. This manuscript is well-organized with reliable results and delicate illustrations, which will have a significant impact on the research fields of catalysis and synthetic chemistry. Therefore, I recommend its acceptance in Nat. Commun. after addressing the following minor issues.

Response: Thanks for the comments. We have addressed the concerns in the following point-by-point response.

Comment 1: The superiority of this reaction route, in comparison with the conventional synthesis methods, should be indicated, including the synthesis efficiency, reaction mechanism and costs.

Response: Thanks for the comments. We have supplemented a table for this issue to fully compare the photocatalysis route for CH₃CONH₂ synthesis with conventional industrial synthesis methods.

Page 7 in Manuscript: Moreover, when compared with the conventional industrial synthesis route (Supplementary Table 1), the photocatalytic co-oxidation route for CH₃CONH₂ synthesis demonstrates significant progress in terms of synthesis conditions, raw material price, and synthesis efficiency.

Supplementary Table 1 | Comparison of the photocatalytic route for the CH₃CONH₂ synthesis with the conventional route for the industrial CH₃CONH₂ synthesis.

	Photocatalytic route	Conventional route
Raw materials	CH ₃ CH ₂ OH + NH ₃	CH ₃ COOH + NH ₃

Reaction temperature	25 °C	200 °C
Synthesis efficiency	97.67% ± 1.67% (N-selectivity) 95.60%±0.22% (C-selectivity)	87%~90%
Reaction mechanism	$\text{CH}_3\text{CH}_2\text{OH} + \text{NH}_3 \rightarrow \text{CH}_3\text{CONH}_2 + 2\text{H}_2$	$\text{CH}_3\text{COOH} + \text{NH}_3 \rightarrow \text{CH}_3\text{COONH}_4$ $\text{CH}_3\text{COONH}_4 \rightarrow \text{CH}_3\text{CONH}_2 + \text{H}_2\text{O}$
Cost (Price in December 2023)	Ethanol: 586 USD/MT Ammonia: 760 USD/MT	Acetic Acid: 595 USD/MT Ammonia: 760 USD/MT

Comment 2: *In a heterogeneous system, a decrease in mass transfer and light utilization efficiencies is inevitable. It should be illustrated how the high reaction rate is ensured at the light-irradiated catalysis interface.*

Response: Thanks for the comments. Regarding the improvement of mass transfer efficiency and light utilization, we have added respective descriptions in the Manuscript.

Page 16 in Manuscript: For the overall photocatalytic synthesis pathway of CH_3CONH_2 , optimizing the interaction between the initial reactive species and the photogenerated charge carriers is of crucial importance to achieve carrier-driven radical generation. By fully leveraging the radical characteristics of a long lifetime and migration distance, radical-mediated photocatalytic reactions can proceed effectively, which ensures the high efficiency of mass transfer and the sufficient utilization of light energy, thereby achieving a significant enhancement of the redox efficiency.

Comment 3: *More detailed reaction parameters for the route screening (Table 1) should be appended to guarantee its reproducibility.*

Response: Thanks for the helpful suggestions. The respective description has been added in the Supplementary Information.

Supplementary Note 1 | Reaction route screening experiment.

300.00 mg L⁻¹ of different N-source (NH_3 , NO_2^- or NO_3^-) and 35 mmol different C-source (CH_3OH , $\text{CH}_3\text{CH}_2\text{OH}$, HCOOH , CH_3OOH , CH_3COONa) were added into 50.00

mL acetonitrile, in which arranged 13 routes (Table 1 in the Manuscript). The 300.00 mg L⁻¹ NH₃ is 0.5 vol.% of NH₃ gas (50 mL min⁻¹) continuously injected into the reactor. The 300.00 mg L⁻¹ NO₂⁻ is 1.50 mL of NO₂⁻ (100000.00 mg L⁻¹) dissolved in 50.00 mL of reaction solution. The 300.00 mg L⁻¹ NO₃⁻ is 0.15 mL of NO₃⁻ (10000.00 mg L⁻¹) dissolved in 50.00 mL of reaction solution.

Comment 4: A redox route of NH₃ oxidation and CH₃COOH reduction is conducted for route screening. It is confusing that only trace acetamide is fabricated since NH₃ and CH₃COOH are typical reactants for industrial acetamide synthesis. I assume the reason may be the proceeding neutral reaction in the liquid phase, which decreases the activity of NH₃ molecules and delivers CH₃COONH₄ as the side product. I recommend replacing CH₃COOH with CH₃COONa as the reductive reactant to test the catalytic efficiency.

Response: Thanks for the comments. According to your suggestion, we have replaced CH₃COOH with CH₃CONa as the reductive reactant to test the catalytic efficiency, in which only trace CH₃CONH₂ is detected. The respective result has been added in the modified Manuscript.

Table 1. List of screening experiments of different photocatalytic redox routes for amides synthesis.

C-source	N-source	Product	Synthesis rate ($\mu\text{mol h}^{-1}$) ^a
CH ₃ OH	NH ₃	HCONH ₂	51.99
CH ₃ OH	NO ₂ ⁻	HCONH ₂	40.44
CH ₃ OH	NO ₃ ⁻	HCONH ₂	41.21
HCOOH	NH ₃	HCONH ₂	0.32
HCOOH	NO ₂ ⁻	HCONH ₂	trace
HCOOH	NO ₃ ⁻	HCONH ₂	0.57
CH ₃ CH ₂ OH	NH ₃	CH ₃ CONH ₂	109.24
CH ₃ CH ₂ OH	NO ₂ ⁻	CH ₃ CONH ₂	23.41
CH ₃ CH ₂ OH	NO ₃ ⁻	CH ₃ CONH ₂	16.01
CH ₃ COOH	NH ₃	CH ₃ CONH ₂	trace
CH ₃ COONa	NH ₃	CH ₃ CONH ₂	trace
CH ₃ COOH	NO ₂ ⁻	CH ₃ CONH ₂	6.55
CH ₃ COOH	NO ₃ ⁻	CH ₃ CONH ₂	trace

^a The respective standard curves of HCONH₂, CH₃CONH₂ and CO(NH₂)₂ (Supplementary Figures. 1-3) are provided in the Supplementary Information.

Comment 5: *Since the solid acetamide product is obtained, its NMR purity should be tested and appended in the revised manuscript.*

Response: Thanks for the comments. We carried out the ^1H NMR test on the products obtained from the long-period experiments after rotary evaporation, and the obtained ^1H NMR spectrum is as follows, there are only peaks ascribed to $-\text{NH}_2$ (δ 7.30 and δ 6.70), $-\text{CH}_3$ (δ 1.76) and H_2O (δ 3.40), so it can be concluded that the solid products are pure CH_3CONH_2 . The respective description has been added in the modified Manuscript and Supplementary Information.

Page 8 in Manuscript: The collected solid sample, with rotary evaporation for its recovery, is verified as CH_3CONH_2 by the XRD and nuclear magnetic resonance (^1H NMR) technology, consistent with the crystallography open database (COD #9007523, Figure 2f) and chemical shift of the hydrogen atom (Supplementary Figure 26) respectively.

Supplementary Figure 26 | Nuclear magnetic resonance (^1H NMR) result for the products after the long-term stability test.

Comment 6: *The C- and N-based intermediates are accumulated for C-N coupling.*

Then how to exclude the possibility of C-C or N-N coupling?

Response: Thanks for the comments. This concern is addressed from the perspective of the reaction kinetics. C-intermediates are long-lived and highly concentrated, but reactively inert. However, N-intermediates have shorter lifetimes, lower concentrations, and tend to quench each other, but are highly reactive. In the case of C-C coupling, the reaction rate will be extremely slow. However, in the case of N-N coupling the reaction selectivity is poor and the product concentration is low. Therefore, we couple the long-lived intermediate with the short-lived intermediate to ensure the rapid generation of products with high selectivity. In our experiment, we have detected no product of C-C or N-N coupling compound. Accordingly, we describe this issue in the manuscript.

Page 15 in Manuscript: The coupling of long-lived C-intermediate and short-lived N-intermediate ensures high selectivity and high yields of CH₃CONH₂.

Comment 7: the NH₂ species are detected by both FTIR and EPR measurements. What is the difference between the -NH₂ in FTIR and •NH₂ in EPR? How to distinguish these similar intermediates and how to identify their specific roles?

Response: Thanks for the comments. The main difference between -NH₂ and •NH₂ is their electronic structure and existence form. •NH₂ represents a free radical, which contains unpaired electrons and can exist independently in the medium (gas or liquid). -NH₂ represents an amino functional group, with eight electrons around the N atom. Besides, -NH₂ cannot exist alone and is often used as a functional group of a compound or attached to a catalyst surface atom. Separate detection of •NH₂ and -NH₂ is possible by different methods (EPR and FT-IR techniques). When -NH₂ species diffuse to the solution, they will transform to •NH₂ and participate in subsequent C-N coupling processes. The respective description has been added in the modified Manuscript.

Page 14 in Manuscript: The DMPO-¹⁴NH₂ and DMPO-¹⁵NH₂ are detected without O₂ provision respectively (Figure 4h, Supplementary Figures 56 and 57, and Supplementary Note 23),^{46,47} manifesting that h⁺ is the dominant driving force for the mild oxidation of NH₃ for •NH₂ generation. By combining the FT-IR and EPR spectra (Figures 3c and 3g, Figure 4h), it is indicated that the absorbed -NH₂ is oxidized by h⁺

to produce $\bullet\text{NH}_2$, which is involved in subsequent C-N coupling processes.

Comment 8: Only the effect of O₂ activation on the intermediate generation is illustrated by EPR. However, the key contribution of O₂ activation to the regulation of the reaction coordinates should be investigated, which may lead to different selectivity for the final products.

Response: Thanks for the comments. We conducted additional investigations on the product distribution at different O₂ concentrations. At 0% O₂ concentrations, the final products are mainly composed of N₂ and CH₃CHO. At the 100% O₂ concentration, NH₃ and CH₃CH₂OH will be peroxidized, causing a significant decrease in the selectivity of CH₃CONH₂, the byproducts mainly consist of NO₃⁻, NO₂⁻, CH₃CHO and CH₃COOH. Therefore, only at 75% O₂ concentration will maximize the selectivity of CH₃CONH₂. The respective results are appended below (Supplementary Figure 41):

Supplementary Figure 41 | Products concentration of photocatalytic C-N coupling at different oxygen concentrations. (a) and (b) under the O₂ proportion of 0%; (c) and (d) under the O₂ proportion of 75%; (e) and (f) under the O₂ proportion of 100%.

Comment 9: The structural formula of CH₃•CHOH (Fig. S37) is not consistent with those in the main manuscript. Also, I believe the trapping agent should be TEMPO rather than TEMP.

Response: Thanks for the comments. We have corrected this mistake in Supplementary Information.

Supplementary Figure 44 | High-resolution mass spectra (HR-MS) result for the TEMPO trapped CH₃•CHOH.

Response to Referee 4:

General Comment: In this manuscript, the authors present a method for synthesizing acetamide from various nitrogen and carbon sources. I appreciate the extensive number of experiments conducted in this study, and the authors claim very high selectivity and activity of the product. However, there are many unclear points regarding the results and methodology, making it unsuitable for publication in Nature Communications. Please see my comments below.

Response: Thanks for the comments. We have addressed the concerns in the following point-by-point response.

Comment 1: The intensity (y-value) in the y-axis of the calibration experiments/curves in Figure S1 should be provided and started from 0. Why the b value (83840.0 a.u?) in the equation of the standard curve is very large? Does it mean that at zero formamide concentration, the signals of this substance in the HPLC are already very substantially high? Why is that? Similar phenomena can be seen in the Figure S2 and S3. How the intensity in the y-axis in these standard curves was measured? It is the height of the peak or the area of the peak? A similar query is raised for Figure S9.

Response: Thanks for the comments. According to your suggestion, we have responded to your suggestions and revised the corresponding figures point-by-point below.

● *The intensity (y-value) in the y-axis of the calibration experiments/curves in Figure S1 should be provided and started from 0.*

The intensity of the y-value is provided and started from 0 in the revised Supplementary Figure 1.

● *Why the b value (83840.0 a.u?) in the equation of the standard curve is very large? Does it mean that at zero formamide concentration, the signals of this substance in the HPLC are already very substantially high? Why is that? Similar phenomena can be seen in the Figure S2 and S3.*

As for the b value (83480.0 a. u.) in Supplementary Figure 1, the value, though

present, is minimal when compared to the actual concentration, which stands at 100.57 ppm with an intensity of 409743 a.u. This highlights its negligible impact in the overall context of the experiment.

● *How the intensity in the y-axis in these standard curves was measured? It is the height of the peak or the area of the peak? A similar query is raised for Figure S9.*

The y-values for the standard curves are derived from the integral of the peak area, a method that ensures the accuracy of the measurements. The y-value at the intercept is significantly lower than the y-value corresponding to the minimum concentration on the curve. This is because, at a formamide concentration of zero, the characteristic peaks for the substance do not manifest in the chromatogram; hence, the peak area integral, and consequently the chromatogram, registers a value of zero, indicating the absence of the substance in question. This approach guarantees that the detection threshold and the baseline of the assay are well-defined and that the measurements are robust against background signals.

In the analysis of the calibration curves, the following points highlight the relationship between the intercept (b value) and the minimum detectable concentrations: **Supplementary Figure 2:** The intercept (b value) is noted at 493,122 arbitrary units (a.u.), while the y-value for a concentration of 52.41 ppm acetamide reaches 1,333,546 a.u. This demonstrates a substantial difference between the baseline and the measurable signal, indicating a clear distinction between background and actual substance presence. **Supplementary Figure 3:** Here, the b value is recorded at 1,058.2 a.u., and for a concentration of 1.02 ppm urea, the y-value is 1,602 a.u. Similar to the first figure, the b value is significantly lower than the y-value at the minimum concentration, ensuring the sensitivity and accuracy of the detection method.

Supplementary Figure 10: This figure, a relabeling of the original Supplementary Figure 9, presents an exception with a b value of -11.7 a.u. and a y-value of 5.5 a.u. for a hydrogen concentration of 0.18 μmol . The negative b value suggests a slight background signal, which is attributed to the low concentration of the hydrogen sample being near the minimum detection limit of the gas chromatography-thermal conductivity detector (GC-TCD) used. Despite this, the actual hydrogen sample

concentrations in the experiment exceeded 0.18 μmol , ensuring that the detection was reliable and the results were not compromised.

Supplementary Figure 1 | Tested standard intensities (a) and standard curves (b) for formamide (HCONH₂) detection by HPLC.

Supplementary Figure 2 | Tested standard intensities (a) and standard curves (b) for acetamide (CH₃CONH₂) detection by HPLC.

Supplementary Figure 3 | Tested standard intensities (a) and standard curves (b) for urea [CO(NH₂)₂] detection by HPLC.

Supplementary Figure 10 | Tested standard intensities (a) and curves (b) for H₂ detection by a GC-TCD.

Comment 2: Why does double catalyst dosage (Figure 2a), from 1 to 2 mg, result in a nearly 8 times increase in production rate? As the Authors proved that P25 does not change throughout the durability test, so by doubling catalyst loading, we can assume that the active surface area is doubled, leading to doubled amounts of products.

Response: Thanks for the valuable suggestions. We appreciate the opportunity to clarify and correct the graphical representation in our study. Here's a refined explanation of the adjustments made to the histograms.

Your suggestion was instrumental in identifying a graphical error in our initial presentation. Typically, when plotting histograms, it's common practice to use sequential integers (1, 2, 3, 4, 5) for the x-axis to ensure neat alignment of the bars. These placeholders are meant to be replaced with the actual values representing catalyst loadings (1.0, 5.0, 10.0, 25.0, 50.0) once the initial layout is set.

The histograms for both unit yield and absolute yield demonstrate an increase in acetamide production with higher catalyst masses. However, it's important to note that the efficiency of catalyst utilization per unit surface area declines as the mass of the catalyst increases. This decline is due to the constraints imposed by the irradiation area

and the volume of the solution, leading to a decrease in unit yield. The interplay between light utilization efficiency and mass transfer efficiency is such that it prevents a direct linear correlation between the rate of reaction and the mass of the catalyst.

We have since corrected the figures to accurately reflect these relationships and to ensure that they align with the actual experimental data. The revised figures are presented below for your reference (Figure 2 and Supplementary Figure 8). We believe these corrections provide a clearer and more accurate representation of the experimental outcomes, and we are grateful for the opportunity to enhance the quality of our presentation.

Figure 2. Efficiency evaluation for acetamide photosynthesis: (a) catalyst dosage-dependent unit production rate; CH₃CONH₂ selectivity evaluation regarding the N-(b) and C-sources (c) respectively; (d) Efficiency comparison between different catalytic routes for C-N coupling, including the targets of production rate and selectivity, the corresponding research works are listed and cited in the Supplementary Information (Supplementary Table 1); (e) Long-term stability test. (f) XRD pattern and the image (inset) of the collected CH₃CONH₂ sample with rotary evaporation after the long-term stability test. The respective standard curves for detecting the reaction species using ion chromatography (IC, NH₄⁺, NO₂⁻, NO₃⁻, CH₃COOH and NH₂OH, Supplementary Figures. 31-35) are provided in the Supplementary Information. The error bars in (a-c) were drawn based on the calculated standard error of two parallel tests.

Supplementary Figure 8 | Screening of the optimum catalyst dosage for acetamide synthesis. The error bars were drawn based on two parallel experiments. (a) The unit yield of different catalyst dosages for acetamide. (b) The absolute yield of different catalyst dosages for acetamide.

Comment 3: *In Figure 2b, the legend shows 8 different N-sources, but the data only shows 4 lines. A similar question is asked in Figure 2c. Is it because some of the results are nearly zero?*

Response: Thanks for the comments. We acknowledge that in Figures 2b and 2c, certain data points are minimal or barely detectable, which can lead to them appearing as a single, unified line. To address this and to enhance clarity, we have included a comprehensive breakdown of the specific data and corresponding standard curves in the Supplementary Figures 12-25, along with detailed Source Data files. To further elucidate, we have added a supplementary explanation within the manuscript itself, ensuring that readers have access to a clear and thorough understanding of the data presentation. This additional context aims to provide transparency regarding the data points that may seem obscured due to their minimal values.

Page 7 in manuscript: The concentrations of the other by-products are exceedingly low, resulting in data points in Figure 2b that are so closely aligned they appear to overlap, making individual distinctions hard to discern.

Comment 4: *In Supplementary Note 5, the Authors mentioned that the reaction*

solution was extracted for the CH₃CONH₂ every 10 h for the stability test, but why the data points in Figure 2e do not show at every 10 h?

Response: Thanks for the comments. To enhance readability and streamline the presentation, we have intentionally omitted certain data points in Figure 2e. However, recognizing the importance of comprehensive data representation, we have updated the figure to include all data points for a more exhaustive view. This revised figure is now incorporated into the manuscript, and the full dataset is accessible in the Source Data 2 file, ensuring that all relevant information is available for thorough analysis.

Figure 2. Efficiency evaluation for acetamide photosynthesis: (a) catalyst dosage-dependent unit production rate; CH₃CONH₂ selectivity evaluation regarding the N-(b) and C-sources (c) respectively; (d) Efficiency comparison between different catalytic routes for C-N coupling, including the targets of production rate and selectivity, the corresponding research works are listed and cited in the Supplementary Information (Supplementary Table 1); (e) Long-term stability test. (f) XRD pattern and the image (inset) of the collected CH₃CONH₂ sample with rotary evaporation after the long-term stability test. The respective standard curves for detecting the reaction species using ion chromatography (IC, NH₄⁺, NO₂⁻, NO₃⁻, CH₃COOH and NH₂OH, Supplementary Figures. 31-35) are provided in the Supplementary Information. The error bars in (a-c) were drawn based on the calculated standard error of two parallel tests.

Comment 5: *From the long-term stability test (Figure 2e), we can see that the amount of products linearly increases over time, reaching 1.82 g of acetamide after 300 h, which is equivalent to ~0.03 mole of NH₃. However, the total mole of NH₃ added to the reactor is only ~0.0024 moles (800 mg L⁻¹ NH₃ in CH₃CH₂OH, and only 50 mL of CH₃CH₂OH was used, so the amount of NH₃ is 40 mg). Why it is possible?*

Response: Thanks for the comments. In our long-term experimental setup, the consistent formation of acetamide is a direct result of the regular supplementation of NH₃. Whenever the NH₃ is depleted, we promptly restore it to the system at the precise concentration needed to maintain the reaction's continuity. The specifics of this process have been elaborated in Supplementary Note 6. For your reference, the source data illustrating this pattern are presented below:

Supplementary Note 6 | Details for the long-term stability test.

The total reaction consists of 50.00 mL of CH₃CH₂OH and 800.00 mg L⁻¹ of NH₃ (in CH₃CH₂OH) are introduced as the C- and N-sources respectively. 75% of O₂ (in Ar) was continuously injected. 50.00 mg of P25 was applied as the catalyst. The NH₃ is periodically provided. When the NH₃ is depleted, 800.00 mg L⁻¹ of NH₃ is added to the long-term stability test system. The reaction solution was extracted for the CH₃CONH₂ test every 10 h, in which the catalyst was washed and dried for further testing and characterization. The collected catalyst samples were characterized by X-ray diffraction technology (XRD, Shimadzu XRD-6100) for its crystal stability, scanning electron microscopy (SEM, FEG ESEM XL30) and transmission electron microscopy (TEM, FEI Talos F200S) for its geometric stability respectively.

Source Data: Fig. 2e

Reaction time (h)	Acetamide yield (g)
0	0
10	0.05891
20	0.11939
30	0.18289
40	0.23697
50	0.28967
60	0.36284
70	0.41532
80	0.48465
90	0.55299
100	0.60517
110	0.68499
120	0.73785
130	0.80223
140	0.85646

150	0.92579
160	0.98056
170	1.03091
180	1.08279
190	1.16159
200	1.21448
210	1.25391
220	1.31948
230	1.38358
240	1.43569
250	1.49498
260	1.54768
270	1.61673
280	1.69833
290	1.75066
300	1.81639

Comment 6: From the in situ ATR-FTIR in Figure 3c, f, g, we can see that the amount of C-N coupling is saturated after around 20 min of irradiation, so why in the long-term test the total production keeps increasing linearly over time?

Response: Thanks for the comments. In accordance with the operational mechanism of the in-situ ATR FT-IR, as depicted in Supplementary Figure 36, the spectral signals are derived from species that are adsorbed on the catalyst's surface. We concur that the saturation of C-N coupling species adsorbed on the surface is observable; however, it is important to note that additional coupling species are continuously released into the solution, contributing to the ongoing synthesis of CH₃CONH₂. The detection of CH₃CONH₂ by HPLC captures these species that are present in the solution, not those merely adsorbed on the catalyst's surface. Consequently, this dynamic interaction between the surface and the solution allows for a sustained linear increase in the total production of CH₃CONH₂ over the course of the long-term experiments.

Supplementary Figure 36 | Schematic diagram of *in-situ* liquid ATR FT-IR.

Comment 7: *In Figure 3a, what are the peaks for the spectrum in the dark? Why the C-OH peak is not observed in for this sample in the dark? Similarly, why there is no peak (N-H) for ammonia in Figure 3b in the dark? In Figure 3c, after mixing CH₃CH₂OH and NH₃, why the peaks appear in Figure 3a for EOR does not show in this Figure in the dark?*

Response: Thanks for the comments. As for the dark spectra, the "In dark" curve depicted in Figure 3a represents the adsorption equilibrium, which was determined by calculating the difference between the second-to-last and the final spectral recordings captured by the spectrometer in the absence of illumination. To provide a more comprehensive understanding, the spectral data illustrating the adsorption process as it unfolds over time in the dark have been detailed in the Supporting Information (Supplementary Figures 37 to 39). This additional data offers insight into the dynamic changes occurring on the catalyst surface prior to photoactivation.

In the Supplementary Figure 37a, the C-OH and O-H peaks, which are indicative of adsorption under dark conditions, are observed at 1376 cm⁻¹ and 1043 cm⁻¹, respectively. This data, along with the detailed attribution of IR peaks presented in the Supplementary Table 3, confirms the adsorption of CH₃CH₂OH molecules onto the catalyst surface in the absence of light. Similarly, in Supplementary Figure 38a, the N-H adsorption peak is identified at 3194 cm⁻¹, as documented in Supplementary Table 4. This observation suggests that NH₃ molecules also undergo adsorption on the catalyst

surface under dark conditions. These findings are crucial for understanding the initial states of the reaction before photoactivation proceeds.

During the co-oxidation of $\text{CH}_3\text{CH}_2\text{OH}$ and NH_3 in the dark, as detailed in the Supplementary Figure 39a, the N-H peak is detected at 3044 cm^{-1} and the C-OH peak at 1603 cm^{-1} . These specific peak positions, along with their respective citations, are clearly outlined in Supplementary Table 5. This data provides direct evidence for the co-adsorption of NH_3 and $\text{CH}_3\text{CH}_2\text{OH}$ molecules on the catalyst surface under dark conditions, which is a primary step in the reaction mechanism, leading to the formation of the coupling products.

As to your last comment, we have identified two primary reasons for the observed phenomena. Firstly, the co-oxidation processes of ethanol oxidation reaction (EOR) and ammonia oxidation reaction (AOR) lead to the efficient consumption of intermediate products, thereby preventing the formation of peroxide compounds. Secondly, the overlapping of characteristic peaks in the FT-IR spectra is attributed to their close proximity in the oxidation reaction pathway, which results in the distinct lines observed in the dark conditions.

To clarify these points and prevent any potential confusion, we have made additional explanations in the manuscript. Furthermore, we have updated all FT-IR spectra labels to replace "In dark" with "0 min" for consistency and to accurately represent the time point at which these measurements were taken. These revisions aim to provide a clearer understanding of the experimental observations and the underlying chemical processes.

The corresponding description, figures and tables are revised based on your valuable suggestions.

Page 9 in manuscript: As the individual EOR proceeds (Figure 3a, the spectra for the adsorption equilibrium process in the dark are exhibited in Supplementary Figure 37 and Supplementary Table 3),

Page 9 in manuscript: the consumption of NH_3 (N-H at 3099 cm^{-1})³⁷ yields the intermediated products of $-\text{NH}_2$ (1090 cm^{-1})³⁸ and NH_2OH (1268 cm^{-1})⁸ in the individual AOR (Figure 3b, the spectra for the adsorption equilibrium process in the dark are exhibited in Supplementary Figure 38 and Supplementary Table 4),

Page 9 in manuscript: Most importantly, as illustrated in the combined co-oxidation reactions of EOR and AOR (Figure 3c, the spectra for the adsorption equilibrium process in the dark are exhibited in Supplementary Figure 39 and Supplementary Table 5),

Figure 3. *In-situ* ATR-FTIR investigation for revealing the reaction coordinates: Time-dependent IR signals for the individual ethanol oxidation reaction (EOR, a), ammonia oxidation reaction (AOR, b) and combined EOR and AOR (c) respectively; Normalized results of the IR signals of COO^- (d), NO_3^- (e), $\text{C}=\text{O}$ (f) and $-\text{NH}_2$ (g) respectively. The full spectra of the adsorption and photocatalysis process are provided in the Supplementary Information for individual EOR (Supplementary Figure 37), individual AOR (Supplementary Figure 38) and combined EOR and AOR (Supplementary Figure 39) respectively.

Supplementary Figure 37 | *In-situ* ATR-FTIR results for the adsorption in dark (a) and photocatalysis (b) process of individual EOR.

Supplementary Figure 38 | *In-situ* ATR-FTIR results for the adsorption in dark (a) and photocatalysis (b) process of individual AOR.

Supplementary Figure 39 | *In-situ* ATR-FTIR results for the adsorption in dark (a) and photocatalysis (b) process of combined EOR and AOR.

Supplementary Table 3 | Assignments of the IR bands in the absorption and photocatalysis process of individual EOR.

Wavenumber, cm^{-1}	Assignment	Refs.
1657	C-O	8
1603, 1376 and 1364	C-OH	9
1519	C=O	10
1420	COO^-	11
1043	O-H	9

Supplementary Table 4 | Assignments of the IR bands in the absorption and photocatalysis process of individual AOR.

Wavenumber, cm^{-1}	Assignment	Refs.
------------	-------

3194, 3099 and 1421	N-H	12
1755	N-O	13
1626	NO ₃ ⁻	14
1530	NO ₂ ⁻	15
1268 and 1244	NH ₂ OH	16
1090	-NH ₂	17

Supplementary Table 5 | Assignments of the IR bands in the absorption and photocatalysis process of combined EOR and AOR.

Wavenumber, cm ⁻¹	Assignment	Refs.
1603 and 1364	C-OH	9
1544	C=O	10
1416	C-N	18
3087 and 3044	N-H	12
1244	NH ₂ OH	17
1159 and 1090	-NH ₂	19

Comment 8: It is interesting when the data in Figure 3a-c in the dark and after 10 min of irradiation are substantially different. If the reactions occur, should the products/intermediates appear gradually rather than substantially change? So we should expect a gradual change between the dark and 10 min irradiation. Can it be revealed if the spectra for the shorter irradiation time are obtained?

Response: Thanks for the comments. In response to the inquiry about shorter irradiation times, we have indeed collected data that illustrates the progressive buildup of products over time. The decision to use a broader time gradient in Figure 3a-3c was strategic to highlight the contrast in product accumulation before and after the initiation of illumination. Following your recommendation, we have adjusted the infrared spectra presented in Supplementary Figures 37b, 38b, and 39b. By reducing the sampling interval, we are now able to depict the incremental accumulation of products from the individual ammonia oxidation reaction (AOR), ethanol oxidation reaction (EOR), and their combined process (AOR+EOR) more clearly. The revised figures, which better capture the subtle changes over a shorter timeframe, are displayed below for your review:

Supplementary Figure 37 | *In-situ* ATR-FTIR results for the adsorption in dark (a) and photocatalysis (b) process of individual EOR.

Supplementary Figure 38 | *In-situ* ATR-FTIR results for the adsorption in dark (a) and photocatalysis (b) process of individual AOR.

Supplementary Figure 39 | *In-situ* ATR-FTIR results for the adsorption in dark (a) and photocatalysis (b) process of combined EOR and AOR.

Comment 9: Another critical point is that the authors compare their work with

different studies in the literature (Fig 2d and Table S1). However, there are too many differences between the methodologies used to collect data, so I don't believe these comparisons are accurate. For example, when electrocatalysis is used, production rates can depend on applied currents, potentials, electrolytes, and cell configurations. It is also difficult to justify comparisons between photocatalysis and electrocatalysis. In photocatalysis, light intensity is the most important factor, and I don't think the light intensity is reported in this work. If the light intensity differs from that in other studies, then there is no valid basis for comparison.

Response: Thanks for the comments. For your first suggestion, we have removed all references to electrocatalytic C-N coupling efficiencies from our comparison. This decision was made to prevent any potential bias or unfairness in comparing the distinct processes of electrocatalysis and photocatalysis, ensuring that our analysis focuses solely on relevant and comparable data.

As for the concern of photocatalysis, given that light intensity is frequently not specified in the literature on photocatalysis, we have incorporated the light source as a critical evaluation parameter for comparing the performance of photocatalytic C-N coupling reactions. This adjustment ensures a more standardized and transparent comparison of different studies. The manuscript and the Supplementary Table have been updated to reflect these changes, providing a clearer and more comprehensive basis for the evaluation of photocatalytic efficiency.

Page 8 in Manuscript: In comparison with the other reported routes for photocatalytic C-N coupling (Figure 2d and Supplementary Table 2),

Figure 2. Efficiency evaluation for acetamide photosynthesis

Supplementary Table 2 | Comparison of the electro/photocatalytic C-N coupling efficiency from recently published literature, including the targets of production rate and selectivity (or Faraday efficiency).

Catalyst	Reaction route	C-source	N-source	Light source	C-N coupling product	Production rate (mmol g _{cat} ⁻¹ h ⁻¹)	Product selectivity/ Apparent quantum yield (%)	Refs.
CeO ₂ -500	photocatalysis	CO ₂	N ₂	300 W Xe lamp	Urea	0.01	Not reported	1
Ru-TiO ₂	photocatalysis	CO ₂	N ₂	300 W Xe lamp	Urea	0.025	6.3	2
Ni ₁ -CdS/WO ₃	photocatalysis	CO ₂	N ₂	300 W Xe lamp	Urea	5.2	0.15	3
Pt cluster/TiO ₂	photocatalysis	CH ₃ OH	N ₂	300 W mercury lamp	Urea	0.11	97.29	4
Bi-TiO ₂	photocatalysis	CH ₃ OH	NH ₃	300 W Xe lamp	Formamide	5.47 ± 0.03	84.8	5
Ba-TiO ₂	photocatalysis	CH ₃ OH	NO ₃ ⁻	365 nm light source	Glycine	0.87	Not reported	6
Ru ₁ /CdS	photocatalysis	LA	NH ₃	300 W Xe lamp	Alanine	41.6	78.6	7
TiO₂ (P25)	photocatalysis	CH₃CH₂OH	NH₃	300 W Xe lamp	Acetamide	105.61±4.86	99.17±0.39	This work

Comment 10: A typo in Supplementary Note 4. The last sentence should be C-source, not N-sources.

Response: Thanks for the comment. This mistake description is now revised in the Supplementary Information.

Supplementary Note 5 | Details for the CH₃CONH₂ selectivity (C) test.

The CH₃CONH₂ selectivity for C-source (S_N) was calculated as follows:

$$S_C = \frac{C_{acetamide} * 100\%}{C_{acetamide} + C_1 + C_2 + C_3 + C_4}$$

Response to the Reviewers' Comments

Reviewer #1 (Remarks to the Author):

All my comments have been addressed, and this manuscript is publishable in the present form.

Response: Thanks for your kind comments and positive recommendation to accept this manuscript.

Reviewer #2 (Remarks to the Author):

I am satisfied with those modifications, I recommend accept this manuscript with further revision.

Response: Thanks for your kind comments and positive recommendation to accept this manuscript.

Reviewer #3 (Remarks to the Author):

The authors reported a new photocatalytic radical addition route for gram-scale acetamide synthesis, which is interesting. Meanwhile, the authors revised the manuscript well and addressed most of the questions. Therefore, I would suggest acceptance in Nat Commun.

Response: Thanks for your kind comments and positive recommendation to accept this manuscript.

Reviewer #4 (Remarks to the Author):

Thank you to the authors for their revision. However, given the substantial discrepancies between the data in the initial submission and the revised version, which changes the nature of the methodology used in this work and the reported results, I do not believe this manuscript meets the standards for publication in Nature Communications.

Response: Thanks for these helpful comments. Based on your valuable suggestions, we have tried our best to revise this manuscript and responded to your concerns in a

point-by-point manner, which provided more solid evidence and comprehensive descriptions. We honestly apologize for these deficiencies in the previous revision, mainly including our misunderstanding of your comments, and lack of some important details. After carefully reading and referring to your comments, the manuscript has been appropriately revised and polished. We believe we have addressed all of your and the Editor's concerns in this round of revision. Thanks again for re-reviewing this manuscript.

Comment 1: For my comment 2, the catalyst loading for Figure 2a has been updated to 10 times higher than in the initial submission. The authors attributed this to a 'graphical error.' However, such an error could have led to highly misleading results, making it difficult to compare with future papers. More critically, I find the explanation of a 'graphical error' oversimplified, especially given that the authors simply increased the catalyst loading in response. This raises concerns about the accuracy of the reported catalyst amounts. Even with the revised data, increasing the catalyst loading by 5x resulted in a performance improvement of more than 7.5x, which ties back to my previous question regarding the performance increase.

Response: Thanks for the comments. First, we can confirm that the data reported in the revised version is factual and rigorous. As for the concern of the catalyst loading, we provide our explanation and revisions as follows in a point-by-point manner.

(1) Explanation of the simplified "graphical error".

In our initial submission, there were mistakes indeed, regarding the graphical process. Specifically, the mass of catalyst loading is 1, 5, 10, 25 and 50 mg respectively. However, in the preparation of the figure, if we input the above-mentioned values as the X coordinates, the inconsistent space between these columns makes this figure not tidy and exquisite. So, these columns were first listed along the order of 1, 2, 3, 4 and 5 mg for their consistency. Unfortunately, after drawing the figure, we forgot to revise these X coordinates to their original values. We do feel it is an honest mistake, and we sincerely thank you for the careful review to point out it, which helps us improve this

manuscript and avoid making mistakes.

Figure 2. Efficiency evaluation for acetamide photosynthesis

(2) Response to the concern of the efficiency promotion for more than 7.5x when the catalyst loading is elevated by 5x.

In this work, the synthesis of acetamide is dominantly contributed by radical-guided coupling pathways. Hence it is rationally inferred that the non-linear increase in the production rate is induced by the non-linear generation of the key radicals when elevating the catalyst dosage. To further address that, we appended additional *in-situ* EPR measurements to detect the $\text{CH}_3\text{CH}_2\text{O}^\bullet$ radical under different catalyst masses of 1 and 5 mg respectively (Supplementary Figure 47a and 47b). As depicted by the detected results and integrated signals (Supplementary Figure 47c), it is observed that the integral intensity does not elevate linearly with the catalyst mass increase, which may lead to the results of 7.5x promotion for the production rate.

We propose the following reasons to explain this non-linear increase of the radical generation. 1) The actual radical concentration cannot be directly determined since radical annihilation, triggered by the introduction of the DMPO agent, is inevitable when conducting an EPR experiment. 2) Abundant chain reactions would proceed in this radical-included complicated reaction system. Even though these chain reactions cannot be fully revealed by using current EPR technologies, it is predictable that the production rate of acetamide will not be increased linearly by simply elevating the

catalyst dosage.

Supplementary Figure 47 | *In-situ* EPR analysis for detecting $\text{DMPO-CH}_3\text{CH}_2\text{O}^\bullet$ at varying catalyst dosages: (a) 1 mg, (b) 5 mg, and (c) integrated signal intensity comparison for 1 mg and 5 mg.

Besides, by elevating the catalyst dosage, these results of non-linear promotion of catalytic efficiency are widely reported in the research field of heterogeneous photocatalysis (*Nat. Commun.* 2024, **15**, 2422.; *Nat. Commun.* 2023, **14**, 7115.; *Nat. Commun.* 2022, **13**, 2067.; *Nat. Commun.* 2023, **14**, 6891.; *Angew. Chem. Int. Ed.* **2024**, e202405370.; *J. Am. Chem. Soc.* **146**, 11968-11977.; *Nat. Commun.* 2024, **15**, 337.) Hence, the screening of catalyst dosage is imperative to evaluate the optimum reaction rate by introducing the judge unit of $\mu\text{mol g}^{-1} \text{h}^{-1}$, rather than deducing the efficiencies by simply calculating the catalyst dosage. Here are some examples.

Supplementary Figure 23. Relationship between photocatalytic performance and photocatalysts contents. Dependence of the photocatalytic CO_2 reduction rate over different amounts of $\text{E}_7\text{-Cu}_5\text{Au}_1\text{-TiO}_2$.

Cited from Xie, Z., Xu, S., Li, L. *et al.* Well-defined diatomic catalysis for photosynthesis of C_2H_4 from CO_2 . *Nat. Commun.* 2024, **15**, 2422.

Supplementary Figure 39. The effect of different catalyst concentrations of Ni_{SAPs}-PuCN on the photocatalytic H₂O₂ activity in this work ($\lambda \geq 420$ nm, 60 mW cm⁻²; 30 ml pure water; 25 °C).

Cited from Zhang, X., Su, H., Cui, P. *et al.* Developing Ni single-atom sites in carbon nitride for efficient photocatalytic H₂O₂ production. *Nat. Commun.* 2023, **14**, 7115.

Supplementary Figure 15. Effect of photocatalyst and co-catalyst dosing on photocatalytic hydrogen production. (a) Effect of different photocatalyst dosage on hydrogen evolution performance (Pt: 2 wt. %, light intensity: ~500 mW cm⁻²). (b) Effect of different co-catalytic doses on hydrogen production performance. The mass of PTA photocatalyst is 7 mg.

Cited from Guo, Y., Zhou, Q., Nan, J. *et al.* Perylenetetracarboxylic acid nanosheets with internal electric fields and anisotropic charge migration for photocatalytic hydrogen evolution. *Nat. Commun.* 2022, **13**, 2067.

Supplementary Figure S37. H₂O₂ generation rates of photocatalytic half reaction of TT-CTP and TTH-CTP under different photocatalyst concentrations.

Cited from Wang, S., Xie, Z., Zhu, D. *et al.* Efficient photocatalytic production of hydrogen peroxide using dispersible and photoactive porous polymers. *Nat. Commun.* 2023, **14**, 6891.

Figure S12. Optimization of reaction conditions for glycine formation. Reaction conditions: anatase TiO₂ nanoparticles as catalyst, 1 mmol nitrate, methanol as carbon source, 10 mL ultra-pure water, t = 4 h, T = 50 °C, 1 bar Ar, and 365 nm LED irradiation.

Cited from Li, P., Zhao, W., Wang, K., Wang, T. & Zhang, B. Photocatalytic Synthesis of Glycine from Methanol and Nitrate. *Angew. Chem. Int. Ed.* **2024**, e202405370.

Figure S16. The conversion performance of Fe(III)@ACN as a function of dosage of C₃N₄.

During the sol-gel preparation process, the concentration of SA remained constant at 100 mg/5 mL, while the mass of C₃N₄ was varied to obtain the corresponding activity data.

Cited from He, C. *et al.* Photocatalytic Conversion of Methane to Ethanol at a Three-Phase Interface with Concentration-Matched Hydroxyl and Methyl Radicals. *J. Am. Chem. Soc.* 2024, **146**, 11968-11977.

Supplementary Fig. 15. **Photocatalyst for the AQY in overall water splitting performance.** **a** Schematic illustration of AQY calculated following one-step excitation process [17]; **b** The overall water-splitting performance of D-O-ZIS/Pt/CoO_x with different catalyst masses (detail seen in Supplementary Table 4), Pt and CoO_x as cocatalysts with the wt% ratio of Pt:CoO_x of 1:4.

Cited from Xin, X., Li, Y., Zhang, Y. *et al.* Large electronegativity differences between adjacent atomic sites activate and stabilize ZnIn₂S₄ for efficient photocatalytic overall water splitting. *Nat. Commun.* 2024, **15**, 337.

Comment 2: For my comment 5, the authors have now added the following sentence to the Supporting Information: 'The NH₃ is periodically provided. When the NH₃ is depleted, 800.00 mg L⁻¹ of NH₃ is added to the long-term stability test system.' However, I question why this critical detail was omitted from the initial submission. Without the periodic addition of NH₃, the results of this study could have been drastically different, potentially misleading readers upon publication. Even after

adding this statement, it remains unclear when NH₃ was added. The authors only mention it was added when 'depleted,' but when exactly was that? How many times was NH₃ added, and what was the total amount added? This would directly affect the final production yield and should be clarified.

Response: Thanks for the kind comments to help us demonstrate the experimental details. The corresponding details have been emphasized again in the revised Supporting Information.

(1) Explanation and supplementary for the details of periodical NH₃ provision.

As for the critical detail of the periodical provision of NH₃, we actually provided corresponding descriptions and figures in the Manuscript at our first submission and revision, which aims to clarify the corresponding procedure and necessity. Here are some examples we found in the manuscript.

Abstract: Sufficient CH₃CH₂OH is provided to accumulate the stable intermediate (CH₃CHO). Meanwhile, the limited NH₃ concentration ensures the controllable generation and fast addition of the transient radical (\bullet NH₂) on CH₃CHO.

Page 6 in Manuscript: Based on these results, the recovery of the reaction rate and maintaining of selectivity are expected by conducting cycled experiments through periodical input of limited NH₃.

Page 7 in Manuscript: Under the limited provision of NH₃ (100.00 mg L⁻¹), a near-complete conversion of NH₃ (97.67%±1.67%) for CH₃CONH₂ production is accomplished under the 60 mins' light irradiation, presenting a superior CH₃CONH₂ selectivity (N) of 99.17%±0.39%.

Page 8 in Manuscript: Moreover, the as-constructed reaction system has been continuously operated with sufficient CH₃CH₂OH provision for 300 h, in which periodic NH₃ conversion for stable CH₃CONH₂ production is maintained, delivering a gram scale yield of CH₃CONH₂ (1.82 g, Figure 2e).

Page 8 in Manuscript: Meanwhile, the limited and periodic provision of NH₃ leads to the controllable generation and fast addition of transient N-intermediate (\bullet NH₂) on stable CH₃CHO, in which the rapid consumption of \bullet NH₂ for C-N coupling avoids its

peroxidation for side product generation.

Page 10 in Manuscript: On the contrary, the limited provision of NH₃ yields -NH₂ at the first 20 mins. Then -NH₂ is consumed along with the decrease of NH₃ concentration, which follows the efficiency test result (Figure 2b).

Figure 1.

Figure 4k.

In this revision, to fully explain these details for clarity and reproduction, the corresponding details and raw data are provided according to your suggestions, which guarantees authenticity.

Supplementary Note 6 | Details for the long-term stability test.

The total reaction consists of 50.00 mL of CH₃CH₂OH and 800.00 mg L⁻¹ of NH₃ (in CH₃CH₂OH) are introduced as the C- and N-sources respectively. 75% of O₂ (in Ar) was continuously injected. 50.00 mg of P25 was applied as the catalyst. The NH₃ will be a periodic provision. At intervals of approximately 20 hours, when the concentration of NH₃ approaches depletion (decreasing below 100.00 mg L⁻¹), an appropriate amount of NH₃ (35.00-40.00 mg) is added to the reaction system to restore the NH₃ concentration to a level of c. a. 800.00 mg L⁻¹. NH₃ was added 15 times in total within 300 hours, and the total amount of NH₃ added was 560.00 mg. The reaction solution was extracted for the CH₃CONH₂ test every 10 h, in which the catalyst was washed and dried for further testing and characterization. The collected catalyst samples were characterized by X-ray diffraction technology (XRD, Shimadzu XRD-6100) for its

crystal stability, scanning electron microscopy (SEM, FEG ESEM XL30) and transmission electron microscopy (TEM, FEI Talos F200S) for its geometric stability respectively.

Supplementary Table 1 | Details for NH₃ periodic provision in the long-term stability test.

Reaction time (h)	Remaining concentration of NH₃ before periodic addition (mg L⁻¹)	NH₃ periodic addition (mg)	Concentration of NH₃ after periodic addition (mg L⁻¹)	Acetamide yield (g)
0	800.00	40.00	800.00	0
10	405.76	0	/	0.05891
20	82.62	36.00	800.07	0.11939
30	413.41	0	/	0.18289
40	90.55	36.00	809.21	0.23697
50	409.57	0	/	0.28967
60	64.77	38.00	801.06	0.36284
70	418.11	0	/	0.41532
80	67.73	37.00	805.17	0.48465
90	399.32	0	/	0.55299
100	70.81	38.00	812.65	0.60517
110	354.51	0	/	0.68499
120	47.85	39.00	819.74	0.73785
130	400.61	0	/	0.80223
140	71.57	36.00	798.44	0.85646
150	360.56	0	/	0.92579
160	49.02	38.00	797.95	0.98056
170	458.09	0	/	1.03091
180	95.55	35.00	805.01	1.08279
190	370.43	0	/	1.16159
200	74.97	40.00	804.42	1.21448
210	400.42	0	/	1.25391
220	90.17	35.00	800.75	1.31948
230	371.71	0	/	1.38358
240	86.38	36.00	806.38	1.43569
250	435.82	0	/	1.49498
260	105.42	36.00	825.38	1.54768
270	407.00	0	/	1.61673
280	76.21	40.00	796.35	1.69833
290	467.34	0	/	1.75066
300	72.29	0	/	1.81639

Note: NH_3 was added every 20 h, with the amount of 35.00-40.00 mg per addition (to control the NH_3 concentration at $\sim 800.00 \text{ mg L}^{-1}$). NH_3 was added 15 times in total within 300 h, and the total amount of NH_3 added was 560.00 mg.

(2) *Necessity for the periodical provision of NH_3 .*

During the comprehensive experimental process for catalytic efficiency evaluation, it is observed that the excess provision of NH_3 always leads to its per-oxidation, which generates higher-valence products such as N_2 , NO_x and NO_x^- . Hence, to precisely regulate the selective NH_3 oxidation for $\bullet\text{NH}_2$ generation and coupling, the limited and periodical provision of NH_3 in the long-term tests is necessary. Here are some examples in the manuscript to clarify the selectivity of NH_3 oxidation based on the NH_3 provision.

Figure 3b: The absence of $\text{CH}_3\text{CH}_2\text{OH}$ provision in the individual AOR leads to the promoted interaction between NH_3 with excess h^+ (namely, the NH_3 oxidation is intensified), which produces some per-oxidized products, such as NO (1755 cm^{-1}), NO_2^- (1530 cm^{-1}) and NO_3^- (1626 cm^{-1}).

Page 7 in Manuscript: Under the limited provision of NH_3 (100.00 mg L^{-1}), a near-complete conversion of NH_3 ($97.67\% \pm 1.67\%$) for CH_3CONH_2 production is accomplished under the 60 mins' light irradiation, presenting a superior CH_3CONH_2 selectivity (N) of $99.17\% \pm 0.39\%$.

Page 7 in Manuscript: Along with the consumption of NH_3 , the generation of the trace byproduct (CH_3COOH) is observed due to the peroxidation of $\text{CH}_3\text{CH}_2\text{OH}$ under low NH_3 concentration, which again verifies the importance of limited NH_3 provision to guarantee the superior selectivity.

Page 10 in Manuscript: It is again verified that continuous and effective C-N coupling can be accomplished by providing sufficient $\text{CH}_3\text{CH}_2\text{OH}$ and periodically supplying limited NH_3 .

Comment 3: I am unclear on the authors' response to my comment 6. As I initially mentioned, when the FTIR results show no further change, it suggests that either the product concentration has reached saturation or it exceeds the detection limit of the equipment. However, the response provided by the authors seems unclear and does not fully address this point. Additionally, I have another question regarding the in situ measurements: in Figures 3e, f, and g, the Y-axis is labeled as 'Normalized abs.' What exactly is the data normalized to? Is it normalized to the highest data point, the catalyst loading, or something else? Clarifying this is essential for interpreting the presented data."

Response: Thanks for the comment. We are sorry for the previous misunderstanding of your comments. The corresponding explanation has been emphasized to illustrate the procedure of FTIR tests. Also, the normalization method is detailedly described in the revised Supporting Information.

(1) Explanation of the intensity "saturation".

The reason for the intensity "saturation" can be explained by the detection mechanism of the *in-situ* ATR FT-IR. As the appended illustration scheme presents (Supplementary Figure 36), Infrared light penetrates a certain depth into the sample before returning to the surface. During this process, if the sample absorbs in the frequency region of the incident light, the intensity of the reflected light diminishes, resulting in a spectrum similar to that of ordinary transmission absorption. Therefore, it can be used for the qualitative and quantitative analysis of the absorbed chemical composition. A catalysis reaction involves a process of diffusion, adsorption, reaction, desorption, and diffusion again. Infrared spectroscopy can only detect species on the surface of the sample, which are the adsorbed and reactive species. This is because infrared spectroscopy measures the absorption of infrared light by molecules, which is

related to the vibrational modes of the chemical bonds within the molecules. These vibrations are influenced by the molecular structure, and thus, infrared spectroscopy is only sensitive to the species present at the surface of the sample where the adsorption and reaction proceed.

At the first reaction period (such as 0-20 min), the rapid accumulation of reactants, intermediates and products is observed at the catalyst surface. After that, the surface sites of the catalyst are occupied by reactants, products, and intermediates. Therefore, an adsorption-desorption equilibrium is achieved, after which the peak intensity will not continue to increase. However, the achievement of an adsorption-desorption equilibrium on the catalyst surface does not imply that the reaction has ceased. Acetamide is desorbed from the catalyst surface and diffused into the solution, while new acetamide is simultaneously generated on the surface, achieving a cycle of adsorption-desorption. The samples detected in performance tests are also the product concentrations in the solution rather than those on the catalyst surface. Therefore, the “saturation” of the infrared signal does not conflict with the ongoing catalysis process over a long period.

Supplementary Figure 36 | Schematic diagram of *in-situ* liquid ATR FT-IR.

(2) *Details for the normalized FTIR results.*

The data is normalized to the highest data point. By normalization processing, the representation of infrared data is transformed from two-dimensional to one-dimensional, enhancing the comparability of characteristic signals and making differences more distinguishable. For the specific species in the AOR, EOR, and AOR-EOR co-oxidation process, the data of peak heights were respectively extracted for all processes. Between the columns of data, the highest value was set to be 1 and the lowest

value was set to be 0. The rest were correspondingly normalized from 0 to 1. The resulting normalized data were thus described as a function of the IR scanning time.

According to your valuable suggestion, the data processing method of the normalization in Figures 3d-3f is appended in the revised Supporting Information file (Supplementary Note 8). More importantly, the raw data of the normalized results are provided for reproduction (also listed in the file of *Source Data-Fig. 3*).

Page 10 in Manuscript: Then, the signals for the peroxidative byproducts (Figures 3d and 3e) and critical coupling intermediates (Figures 3f and 3g) are normalized for comparison between the individual and combined oxidation reactions. The detailed methods for normalization of the FTIR signals are presented in Supplementary Note 8.
Supplementary Note 8 | *In-situ* ATR-FTIR tests

For the *in-situ* ATR-FTIR test, an INVENIO R FTIR (Bruker) spectrometer equipped with a mercury cadmium telluride (MCT) detector was utilized. All spectra were presented by the absorbance ($-\log(R/R_0)$). The spectral resolution was 4 cm^{-1} , and each curve was scanned for 0.5 min. Before the test, 100.0 uL of catalyst ink (a mixture of 10.00 mg of catalyst, 25.0 uL of Nafion, and 1.00 mL of $\text{CH}_3\text{CH}_2\text{OH}$) was deposited onto the Si crystal and then dried in air. The reaction chamber was filled with a total of 10.00 mL of the reaction solution, and 75% of O_2 (in Ar) was continuously injected into the system. An Xe lamp (Bobei BBZM-1) was applied as the light source, and detection was performed during light irradiation.

The pristine IR signals were normalized to evaluate the species' evolution directly. For the specific species in the AOR process, EOR process, and AOR-EOR co-oxidation process, the data of peak heights were respectively extracted for all processes. Between the columns of data, the highest value was set to be 1 and the lowest value was set to be 0. The rest were correspondingly normalized from 0 to 1. The resulting normalized data were thus described as a function of the IR scanning time.

For instance, for the \$\text{COO}^-\$ species detected at \$1420\text{ cm}^{-1}\$ in the EOR and EOR+AOR, the peak intensities at \$1420\text{ cm}^{-1}\$ were extracted from the spectra collected during the 0-30 min interval. The highest peak intensity within this range was normalized to 1, the

lowest was set to 0, and the remaining values were linearly scaled between 0 and 1. The same normalization process was applied to the NO_3^- species in the AOR and EOR+AOR, represented by the peak at 1626 cm^{-1} , using the data from the 0-30 min interval. The procedure was also followed for the C=O species at 1544 cm^{-1} in the EOR and EOR+AOR, and the $-\text{NH}_2$ species at 1159 cm^{-1} in the AOR and EOR+AOR, where the maximum intensity within the 0-30 min range was set to 1, the minimum to 0, and all other values normalized accordingly. The normalized data obtained from this process are presented below:

Time (min)	COO^-		NO_3^-		C=O		$-\text{NH}_2$	
	EOR	EOR and AOR	AOR	EOR and AOR	EOR	EOR and AOR	EOR	EOR and AOR
0	0	0	0	0	0	0	0	0
5	0.65	0.21	0.26	0.05	0.12	0.45	0.12	0.77
10	0.83	0.14	0.48	0.10	0.22	0.57	0.18	0.82
15	0.81	0.22	0.69	0.06	0.23	0.88	0.25	0.88
20	1.00	0.10	0.99	0.14	0.25	1.00	0.33	1.00
25	0.98	0.23	1.00	0.10	0.26	0.95	0.36	0.65
30	0.95	0.09	0.95	0.10	0.25	0.97	0.35	0.51

Comment 4: For my comments 7 and 8, if you overlay the spectra at 2, 4, 6, 8, and 10 minutes in Figures S37, S38, and S39, there appears to be no significant difference between them. This raises the question of why there is a substantial change from 0 to 2 minutes, but almost no change after 10 minutes, with only a slight variation in peak intensity after 20, 30, and 40 minutes.

Response: Thanks for the comment. These figures can be divided into two reaction processes of adsorption in the dark (Supplementary Figures 37a, 38a and 39a) and photocatalysis under light irradiation (Supplementary Figures 37b, 38b and 39b). Here are the explanations of the intensity variation in these two processes respectively.

(1) Adsorption process in dark.

Under dark conditions, the data curve at 0 min refers to the initial infrared signal on the catalyst surface. To achieve precise tracking of adsorbed N and C substances, we have subtracted the catalyst surface signals as the background, resulting in a relatively flat curve. After the reaction solution is drop-coated on the catalyst surface, the reactive substances rapidly adsorb on the catalyst surface, causing a significant change within the first 2 mins. The characteristic peaks are attributed to reactive species of C and N.

Supplementary Figure 37a: C-OH (1376 cm^{-1}) and O-H (1043 cm^{-1}) species.

Supplementary Figure 38a: N-H (3194 cm^{-1}) species.

Supplementary Figure 39a: C-OH (1603 cm^{-1}) and N-H (3044 cm^{-1}) species.

After 2 min, there is a continuous accumulation process of surface functional groups. Hence, the intensity of the characteristic peaks of the species will increase without the appearance of new substances. As explained in the *Response to Comment 3*, once the adsorption-desorption equilibrium is reached and the sites are fully occupied, the intensity of the characteristic curves tends to stabilize.

(2) Photocatalysis process under light irradiation.

Under light irradiation, the signals at 0 min represent the result of subtracting the adsorption equilibrium curve as the background before turning on the light, which is the initial infrared signals of the reaction catalyst after adsorbing the reactants to equilibrium, and is similarly quite flat with that at the dark condition. Under light exposure, the photocatalytic reaction proceeds, with reactants being rapidly consumed to accumulate intermediates and final products. Hence, there is a significant change within the first 0-2 mins.

Supplementary Figure 37b: decrease of C-OH (1603 cm^{-1} and 1364 cm^{-1}) species, increase of C=O (1519 cm^{-1}), C-O (1657 cm^{-1}) and COO⁻ (1420 cm^{-1}).

Supplementary Figure 38b: decrease of N-H (3099 cm^{-1}) species, increase of -NH₂ (1090 cm^{-1}), N-O (1755 cm^{-1}), NO₂⁻ (1530 cm^{-1}) and NO₃⁻ (1626 cm^{-1}) species.

Supplementary Figure 39b: decrease of N-H (3087 cm^{-1}) and C-OH (1364 cm^{-1}) species, increase of -NH₂ (1090 cm^{-1}) and C-N (1416 cm^{-1}).

After 2 min, the final products continue to accumulate while reactants and some

intermediate products are consumed. Subsequently, the adsorption-desorption of species on the catalyst surface reaches equilibrium, with no significant changes in peak intensity.

Page 67 in the Supporting Information:

These figures (Supplementary Figures 37-39) can be divided into two reaction processes of adsorption in the dark (Supplementary Figures 37a, 38a and 39a) and photocatalysis under light irradiation (Supplementary Figures 37b, 38b and 39b). Here are the explanations of the intensity variation in these two processes respectively.

(1) Adsorption process in dark.

Under dark conditions, the data curve at 0 min refers to the initial infrared signal on the catalyst surface. To achieve precise tracking of adsorbed N and C substances, we have subtracted the catalyst surface signal as the background, resulting in a relatively flat curve. After the reaction is drop-coated on the catalyst surface, the reactive substances rapidly adsorb on the catalyst surface, causing a significant change within the first 0-2 mins. The characteristic peaks are attributed to reactive species such as C and N containing species.

Supplementary Figure 37a: C-OH (1376 cm^{-1}) and O-H (1043 cm^{-1}) species.

Supplementary Figure 38a: N-H (3194 cm^{-1}) species.

Supplementary Figure 39a: C-OH (1603 cm^{-1}) and N-H (3044 cm^{-1}) species.

After 2 min, there is a continuous accumulation process of surface functional groups. Hence, the intensity of the characteristic peaks of the species will increase without the appearance of new substances. As explained in the *Response to Comment 3*, once the adsorption-desorption equilibrium is reached and the sites are fully occupied, the intensity of the characteristic curves tends to stabilize.

(2) Photocatalysis process under light irradiation.

Under light irradiation, the signals at 0 min represent the result of subtracting the adsorption equilibrium curve as the background before turning on the light, which is the initial infrared signal of the reaction catalyst after adsorbing the reactants to equilibrium, which is similarly quite flat with that at the dark condition. Under light exposure, the photocatalytic reaction begins, with reactants being rapidly consumed and

intermediate and final products accumulating, hence there is a significant change within the first 0-2 mins.

Supplementary Figure 37b: decrease of C-OH (1603 cm^{-1} and 1364 cm^{-1}) species, increase of C=O (1519 cm^{-1}), C-O (1657 cm^{-1}) and COO^- (1420 cm^{-1}).

Supplementary Figure 38b: decrease of N-H (3099 cm^{-1}) species, increase of $-\text{NH}_2$ (1090 cm^{-1}), N-O (1755 cm^{-1}), NO_2^- (1530 cm^{-1}) and NO_3^- (1626 cm^{-1}) species.

Supplementary Figure 39b: decrease of N-H (3087 cm^{-1}) and C-OH (1364 cm^{-1}) species, increase of $-\text{NH}_2$ (1090 cm^{-1}) and C-N (1416 cm^{-1}).

After 2 min, the final products continue to accumulate while reactants and some intermediate products are consumed. Subsequently, the adsorption-desorption of species on the catalyst surface reaches equilibrium, with no significant changes in peak intensity.

Comment 5: For my comment 9, I do not feel think that the authors have adequately addressed my question. While the revised version now includes details on the light sources used in both this work and the cited literature, the light intensity is still not reported. Using the same light source, such as a 300 W Xe lamp, can lead to significantly different outcomes depending on the intensity applied. For example, one experiment might use 1 sun (100 mW cm^{-2}), while another could use 2 sun or even 7 sun, which can substantially (and non-linearly) alter the production yield. This critical information must be provided to allow for accurate comparisons and reproducibility.

Response: Thanks for the kind comments for further improvement. We have added a column of light intensity in Supplementary Table 3 for accurate comparison and reproducibility.

Supplementary Table 3 | Comparison of the electro/photocatalytic C-N coupling efficiency from recently published literature, including the targets of production rate and selectivity (or apparent quantum yield).

Catalyst	Reaction route	C-source	N-source	Light source	Light intensity (mW cm ⁻²)	C-N coupling product	Production rate (mmol g _{cat} ⁻¹ h ⁻¹)	Product selectivity/	Refs.
CeO ₂ -500	photocatalysis	CO ₂	N ₂	300 W Xe lamp	Not reported	Urea	0.01	Not reported	1
Ru-TiO ₂	photocatalysis	CO ₂	N ₂	300 W Xe lamp	10	Urea	0.025	6.3	2
Ni ₁ -CdS/WO ₃	photocatalysis	CO ₂	N ₂	300 W Xe lamp	150	Urea	5.2	0.15	3
Pt cluster/TiO ₂	photocatalysis	CH ₃ OH	N ₂	300 W mercury lamp	93	Urea	0.11	97.29	4
Bi-TiO ₂	photocatalysis	CH ₃ OH	NH ₃	300 W Xe lamp	86	Formamide	5.47 ± 0.03	84.8	5
Ba-TiO ₂	photocatalysis	CH ₃ OH	NO ₃ ⁻	365 nm light source	Not reported	Glycine	0.87	Not reported	6
Ru ₁ /CdS	photocatalysis	LA	NH ₃	300 W Xe lamp	Not reported	Alanine	41.6	78.6	7
TiO₂ (P25)	photocatalysis	CH₃CH₂OH	NH₃	300 W Xe lamp	112	Acetamide	105.61±4.86	99.17±0.39	This work